# Atomic structure and defect dynamics of monolayer lead iodide nanodisks with epitaxial alignment on graphene

Sapna Sinha [1], Taishan Zhu[2], Arthur France-Lanord [2], Yuewen Sheng [1], Jeffrey C. Grossman [2], Kyriakos Porfyrakis [3] & Jamie H. Warner[4*]

Lead Iodide ($PbI_2$) is a large bandgap 2D layered material that has potential for semi-conductor applications. However, atomic level study of $PbI_2$ monolayer has been limited due to challenges in obtaining thin crystals. Here, we use liquid exfoliation to produce monolayer $PbI_2$ nanodisks (30-40 nm in diameter and > 99% monolayer purity) and deposit them onto suspended graphene supports to enable atomic structure study of $PbI_2$. Strong epitaxial alignment of $PbI_2$ monolayers with the underlying graphene lattice occurs, leading to a phase shift from the 1 T to 1 H structure to increase the level of commensuration in the two lattice spacings. The fundamental point vacancy and nanopore structures in $PbI_2$ monolayers are directly imaged, showing rapid vacancy migration and self-healing. These results provide a detailed insight into the atomic structure of monolayer $PbI_2$, and the impact of the strong van der Waals interaction with graphene, which has importance for future applications in optoelectronics.

[1] Department of Materials, University of Oxford, 16 Parks Road, Oxford OX1 3PH, UK. [2] Department of Materials Science and Engineering, Massachusetts Institute of Technology, 77 Massachusetts Avenue, Cambridge, MA 02139, USA. [3] Faculty of Engineering and Science, University of Greenwich, Central Avenue, Chatham Maritime, Kent, ME4 4TB, UK. [4] Department of Mechanical Engineering, University of Texas at Austin, 204 Dean Keeton Street, Austin 78712, USA. *email: jamie.warner@austin.utexas.edu

Two-dimensional (2D) materials attract interest because of their unique chemical and physical properties, facilitating the study of novel physics, e.g., trions, valley polarization, etc[1–3]. Although graphene exhibits an exceptionally high carrier mobility ($>10^6$ cm$^2$ V$^{-1}$s$^{-1}$ at 2 K), its zero bandgap poses difficulties for many semiconductor applications[4]. Monolayer transition metal dichalcogenides (TMDs), such as MoS$_2$, can have direct band gaps of ~1.8 eV, but do not exhibit very high charge carrier mobility[5–7]. For Mo- and W-based TMDs, such as MoSe$_2$, WS$_2$, WSe$_2$, etc., the band gaps fall within the range of 1.0–2.0 eV, which is the red to near infrared regions[8,9]. For optoelectronics, blue, green, and ultraviolet light-emitting diodes are also needed for full color displays and cameras. Currently, there are not many experimental studies of 2D materials that can satisfy the demands for the green to UV spectral regions and more research is needed to expand this area.

PbI$_2$ is a layered direct bandgap semiconductor with bandgap of 2.4 eV in its bulk form, whereas its 2D monolayer has an indirect bandgap of ~2.5 eV, with possibilities to tune the bandgap between 1–3 eV[10–13]. PbI$_2$ is frequently used to fabricate organic–inorganic halide perovskite solar cells[14,15], and as a high-energy photon detector material for gamma-rays and X-rays[16–18]. PbI$_2$ has a wide variety of I-Pb-I stacking and this gives rise to more than a dozen polymorphs[19]. However, the thickness of each single layer (0.7 nm), and the distance between each lead and iodide atoms (0.32 nm) is independent of the polytypes[19]. The 2H structure is the most commonly found polytype of three-dimensional PbI$_2$, where each plane of Pb and I atoms are shifted with respect to each other and form overlapping hexagons.

Layered PbI$_2$ has been shown to be excellent candidate for use in optoelectronic applications, photodetectors, and photon detection[20–23]. Ultrathin PbI$_2$ is an interesting system for studying quantum-confinement effects because the exciton–phonon couplings are dependent on the degree of localization of electronic charge[24,25]. Cabana et al.[26] fabricated PbI$_2$ interfaces with carbon nanotubes and studied the change in the density of states of the system. Zhou et al.[10] studied the graphene/PbI$_2$ van der Waals interface and predicted 1.5 eV increase in the visible light absorption capability of the heterostructure as compared to pure 2D PbI$_2$. Recent work on few layer shows PbI$_2$ as a promising candidate for application in the field of ultrafast saturable absorbers[27]. However, research into the atomic structure of monolayer PbI$_2$ has been limited to date because of the challenges in obtaining high-quality monolayer crystals and preparing suitable samples for transmission electron microscopy studies. Further work is needed to reveal the structure and dynamics of edges, point defects, vacancy clusters, and nanopores in PbI$_2$ monolayer and its interaction with other 2D crystals, such as graphene.

Here, we used liquid phase exfoliation (LPE) to isolate PbI$_2$ monolayer flakes from a starting bulk PbI$_2$ powder. Liquid-phase exfoliation is one of the simplest methodologies for producing 2D materials on a large scale[28–30]. In the past decades, various research has been carried out to find suitable solvents based on their interactions with the 2D material to produce suspended 2D monolayer crystals. Surface tension, the Hilderbrand solubility parameter, the Hansen solubility parameter, surface tension components, etc. have been the widely used parameters to screen appropriate solvents[31–35]. Using these parameters, we screened the commonly used solvents and found chloroform (CHCl$_3$) to produce monolayer 2D PbI$_2$ crystals. We used annular dark-field scanning transmission electron microscopy (ADF-STEM) to studying the PbI$_2$ atomic structure, by depositing it from solution onto a suspended graphene support and allowing it to dry[36–38]. The high electron transparency of graphene, enables provides excellent contrast from the Pb and I atoms in ADF-STEM. We report on the fundamental atomic structure, point vacancies, vacancy clusters, vacancy dynamics, edge terminations and edge etching, and the epitaxial interactions with the underlying graphene support.

## Results

**Synthesis.** PbI$_2$ is a unique exception to all the metal halide compounds that show CdI$_2$ structure, in that it has the largest metal halide bond length[39]. As a result, the bonds are not as ionic as that of the other compounds that also show CdI$_2$ crystal structure, such as MgI$_2$, FeBr$_2$, etc. However, it is sufficiently ionic to dissolve into polar solvents, such as dimethylformamide (DMF), n-methyl-2-pyrrolidone (NMP), dimethyl sulfoxide (DMSO), and to some extent, water[40,41]. Recent research on Lewis basicity of solvents, also quantified by Gutmann's donor number—$D_n$, has shown that solvents that solubilize the precursor (PbI$_2$) at a total concentration of 1 M, have higher $D_n$ values of >25 (ref. [42]). Previous results on the liquid-phase exfoliation for other 2D materials have shown that the most efficient solvents have H-bonding components so as to maximize the dispersability of the material in the solution, which is in agreement with the Hansen solubility parameter theory[43,44]. We explored several different solvents and found that chloroform serves as an excellent medium to exfoliate PbI$_2$. CHCl$_3$ has low dipole moment ($\mu = 1.04$), low dielectric constant ($\epsilon_{T = 20\,°C} = 4.8$), low donor number ($D_n = 4$), as well as has the ability to form hydrogen bonds[45–47]. In addition, CHCl$_3$ has a very low boiling point of 61 °C which facilitates in quick and easy removal after solution dropcasting on to substrates.

Figure 1a shows the schematic diagram of the liquid exfoliation experimental procedure. The sonication time is important for controlling the exfoliation of PbI$_2$. Less sonication leaves multilayer PbI$_2$ dispersed in the solution, whereas longer sonication time renders very small flakes or completely destroys the sheets (Supplementary Note 1). Moreover, the supernatant used for dropcasting also plays a role, and only the solution from the top has been used (Supplementary Note 2). Our results show successful exfoliation of mostly monolayer flakes of the PbI$_2$ from its bulk counterpart. Moreover, the dispersed solution was found to be stable for a considerable number of days. Even after a week of the sonication process, we could still observe monolayer flakes dispersed in the solution (Supplementary Note 1). Out of >300 flakes studied in this experiment, we only found 2 flakes with bilayer structure (Supplementary Note 3), whereas all others were monolayers. Some nanoparticles that are observed come from the graphene underneath and not from PbI$_2$ itself (Supplementary Note 4). Figure 1b–d shows the ADF-STEM images of the monolayer PbI$_2$ suspended on monolayer graphene on a TEM grid at various magnifications. The length and width of ~300 flakes was measured (Fig. 1e, f), giving ~42 and 28 nm, respectively, with bigger flakes ranging to >160 nm in size (Fig. 1g).

**1-H structural phase.** The bulk crystal structure of PbI$_2$ has a CdI$_2$ structural type with lead and iodide layers stacked together in ABC instead of ABA stacking. However, in the monolayer form there are two structural phases, namely—1-H phase and 1-T phase. PbI$_2$ has similar structures to the TMDs (MoS$_2$ and WS$_2$), where the metal (Pb) atomic plane is in the middle and the two iodide atomic layers at the top and the bottom. A monolayer from a 2 H PbI$_2$ bulk crystal structure (Fig. 2a) of PbI$_2$ typically has a 1-T phase (Fig. 2b), where the two iodide atoms do not share the same z-axis and are located in an octahedral coordination. Whereas the 1-H phase of monolayer PbI$_2$ (Fig. 2e, f) has two iodide atomic layers on top of each other in a trigonal-prismatic

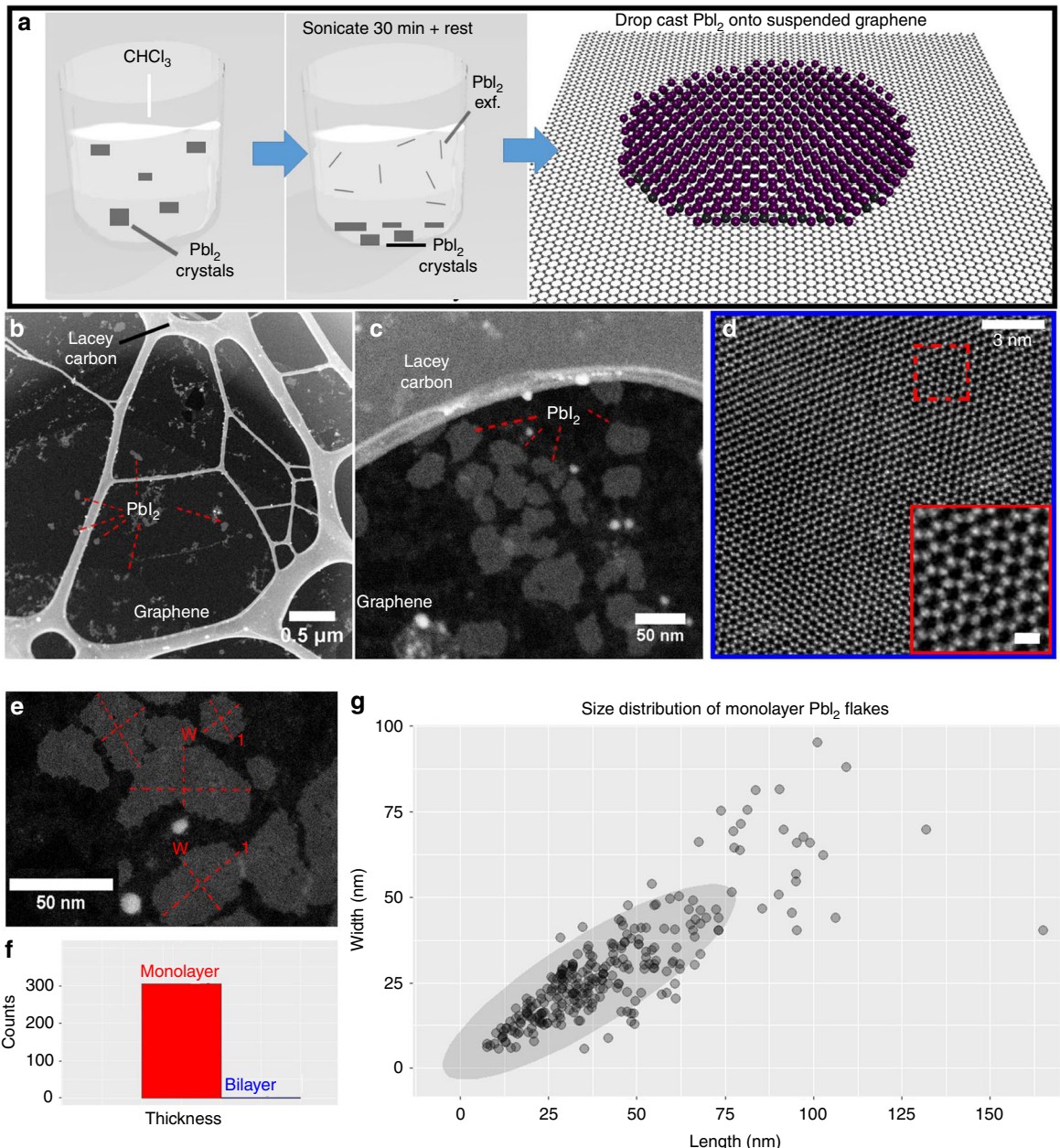

**Fig. 1 Schematic diagram for liquid exfoliation and results. a** Schematic of the LPE synthesis of monolayer PbI$_2$ flakes and deposition on to suspended graphene. **b, c** ADF-STEM image of monolayer PbI$_2$ flakes at different magnifications. **d** Atomic resolution ADF-STEM image of a monolayer PbI$_2$ flake. The inset in the figure shows an atomic resolution image of the structure. The scale bar is 0.5 nm. **e** Typical method for measuring the length(l) and width(w) of the monolayer flake. **f** Histogram showing the thickness distribution of PbI$_2$ flakes. **g** Plot of the size distribution for >300 monolayer flakes. Overlapping sizes of different flakes show darker color for their marker dots. The oval represents the highest congregation of flake sizes in the overall sample distribution.

coordination, as shown in Fig. 2e, f respectively. The position of the Pb and I atoms can be determined by their contrast in ADF-STEM images. Pb has $Z = 82$, I has $Z = 53$, and 2I has $Z = 106$, which means that the 2I positions will have higher contrast than the single Pb. The line profiles (Fig. 2d, h) of the multislice image simulations for 1 T and 1 H PbI$_2$ monolayers give the distance across $1->2$ and $3->4$ in Fig. 2c, g, respectively, for comparison to our experimental data (Fig. 2i–k). The experimental Pb–Pb distance of 1.027 nm and the inter-atomic Pb–2I distances are in excellent agreement with the simulated distances of the 1 H structural type (Fig. 2h). To further validate the case of monolayer 1 H structure, we bombarded the nanodisks with electron beam.

And as seen in Supplementary Note 5, post few minutes of beam exposure, the agglomeration of vacancies lead to hole formation. Previous studies have reported that the initial defect or hole formation occurs layer-wise instead[38,48]. Since, the hole formation revealed no further layers underneath, it is conclusive to say that this is a monolayer structure instead of AA′-stacked bilayer 1 H structure.

The 1 T phase is expected to be more stable than the H phase with total energy 165 meV per unit cell lower[49], unlike the other 2D materials such as MoS$_2$, where the 1 H phase is generally known to be more stable than the 1 T phase[50]. However, all PbI$_2$ flakes suspended on graphene, from smallest 3 nm to larger

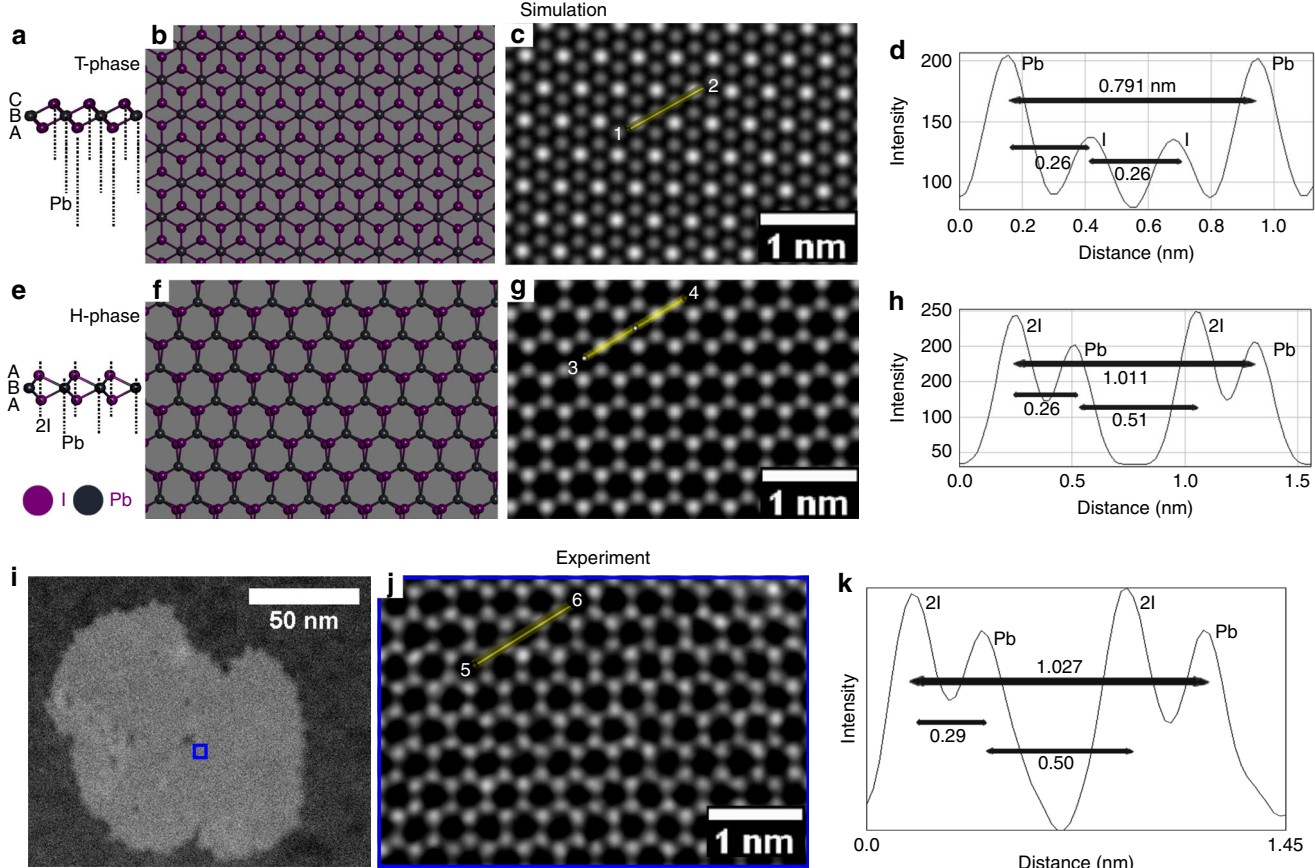

**Fig. 2 Atomic resolution ADF-STEM analysis of monolayer 1 H and 1 T phase. a** Side view of the 1 T atomic structure of PbI₂. **b** Top view of the 1 T atomic model of single layer PbI₂. **c** Multislice ADF-STEM image simulation of the 1 T atomic model in **b**. **d** Line profile of the lead and iodide intensities marked with yellow line in the atomic model in **c**, along the direction 1-2. **e** Side view of 1 H monolayer atomic crystal structure of PbI₂. **f** Top view of the 1 H atomic model of single layer PbI₂. **g** Multislice ADF-STEM image simulation of the 1 H atomic model shown in **f**. **h** Line profile of the lead and iodide intensities marked with yellow line in the atomic model in **g**, along the direction 3-4. **i** Low-magnification ADF-STEM image of a monolayer PbI₂ flake suspended on graphene. **j** Higher magnification of the region indicated by the blue box in **i**, showing the 1 H atomic structure. **k** Line profile taken from region marked by the yellow box in **j**, which corresponds to same position with the region marked in simulation **g**, along the direction 5–6.

hundreds of nanometers in size (Supplementary Note 6), showed 1 H phase, attributed to epitaxial interactions with the underlying graphene support. Chemical vapor deposition (CVD)-grown graphene has grain boundaries with random orientation and hence, we saw that PbI₂ flakes within a region (1–5 μm) showed the same relative crystal orientation relative to the graphene crystal (Supplementary Note 7, Supplementary Note 8)[50,51]. To confirm the case of PbI₂–graphene interactions, we deposited PbI₂ flakes directly on top of lacey carbon TEM grids without graphene, and observed PbI₂ flakes with 1 T structure instead (Supplementary Note 9). We also carried a time-dependent study and concluded that the spatial variations in the PbI₂ lattice occurred during the early stages after deposition on graphene, eventually completely transforming to well-defined 1 H phase over time (Supplementary Note 10). After 1 day, all the PbI₂ flakes on graphene showed well-defined 1 H phase with the same orientations in a given area. To understand why PbI₂ adopts 1 H structure with concomitant alignment to the armchair direction of graphene, we examine the Moire patterns formed between the two crystals (Supplementary Note 11). The best lattice match occurs when the PbI₂ adopts 1 H phase and is aligned to the armchair direction, which agrees with our experimental findings.

Interestingly, we also found that the crystal orientation of PbI₂ on graphene had preferred orientation of PbI₂ zigzag aligned to

graphene armchair direction, with only a small fraction showing the zigzag to zigzag alignments. Figure 3a, f shows the ADF-STEM images and their FFTs of the two different types of orientations of PbI₂ on graphene. Figure 3k–r shows the atomic models of PbI₂ on graphene for the two aligned directions and the multislice image simulation, along with the fast Fourier transform (FFT) results matching the experimental data. To understand why PbI₂ adopts 1 H structure with concomitant alignment in these particular directions of graphene, we employed density functional theory (DFT) calculations to study the structural and energetic stability of the different alignments. We used a short-range relaxed atomic model of PbI₂ on graphene, as shown in Fig. 3s, and rotated it every 5 degrees (Fig. 3s) to find the most energetically stable heterostructure (Fig. 3t). This also confirms our results to how the graphene influences the stability of the PbI₂ and leads to these energetically favorable structure that are also observed in the experimental results.

**Structural defects**. Next, we examine the edge terminations of PbI₂, because edges of 2D materials can influence the electrical, chemical, and catalytic properties due to the different bonding (angles, bond distances, etc.) between atoms[51–54]. Fig. 4a shows the typical edges of two different monolayer PbI₂ flakes, just after being deposited on the graphene. The edges are rough and do not

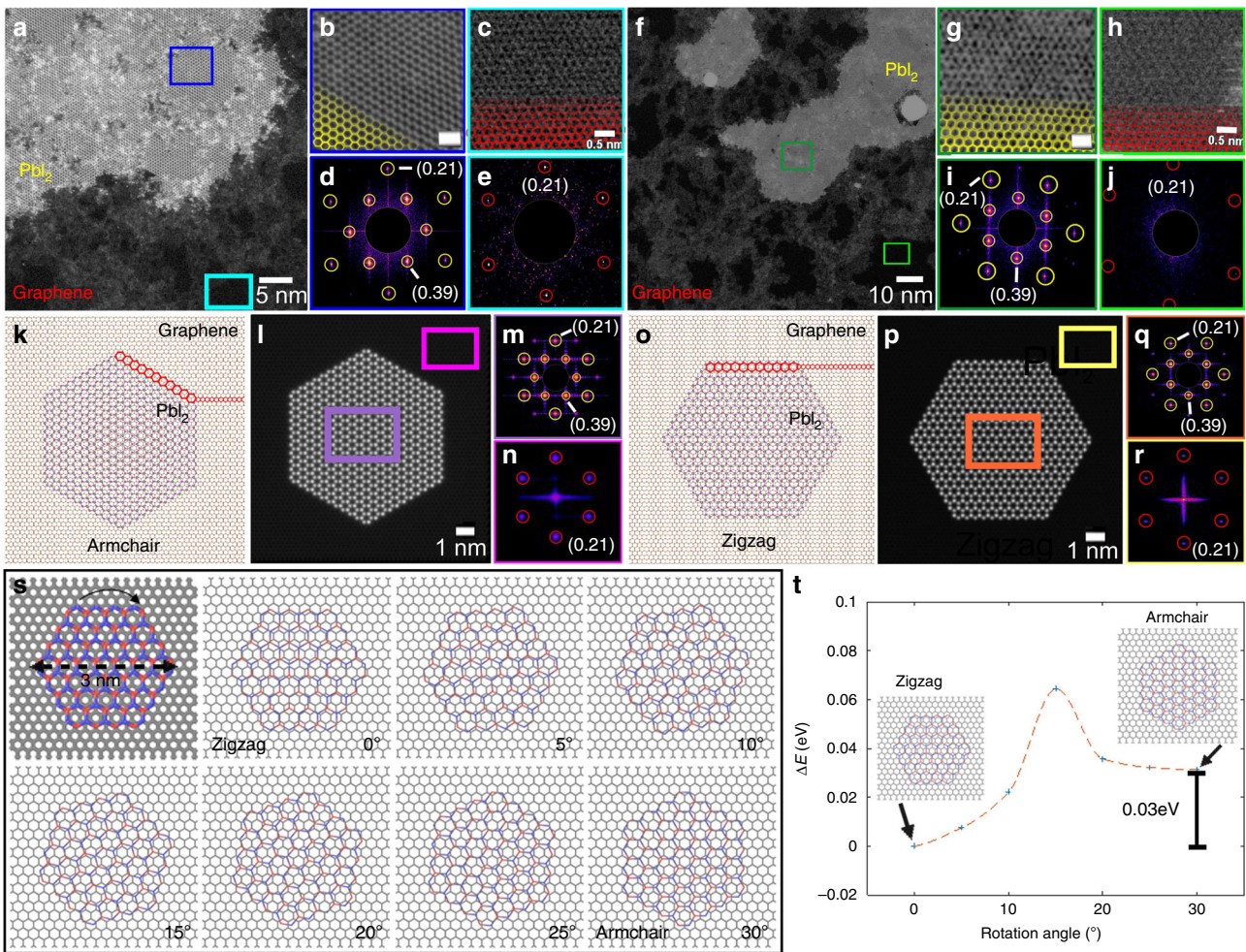

**Fig. 3 Crystal orientation of monolayer PbI₂ on graphene. a** ADF-STEM images of PbI₂ flake suspended on graphene. **b** High-resolution image of PbI₂ crystal annotated in the dark blue box in **a**. Scale bar corresponds to 1 nm. **c** High-resolution image of graphene taken from the cyan box region indicated in **a**. **d** FFT of PbI₂ from the region indicated in **b**. **c** FFT of graphene from the region indicated in **c**. **f** ADF-STEM images of PbI₂ flake suspended on graphene. **g** High-resolution image of PbI₂ crystal indicated by the dark green box in **f**. Scale bar corresponds to 1 nm. **h** High-resolution image of graphene from the light green box in **f**. **i** FFT of PbI₂ from the region indicated in **g**. **h** FFT of graphene from the region indicated in **f**. **k** Atomic model of PbI₂ (1 H) on graphene in the armchair direction. **l** Multislice image simulations based on the atomic model in **k**. **m** FFT of PbI₂ from the region indicated by the purple box in **l**. **n** FFT of graphene from the region indicated by the pink box in **l**. **o** Atomic model of PbI₂ (1 H) on graphene in the armchair direction. **p** Multislice image simulations based on the atomic model in **k**. **q** FFT of PbI₂ from the region indicated by the orange box in **p**. **r** FFT of graphene from the region indicated by the yellow box in **p**. **s** Atomic models showing PbI₂ relative to graphene for **s** DFT-relaxed atomic model of PbI₂ on graphene, rotated every 5 degrees. **t** Energy profile of various orientation angles of the PbI₂ flake on graphene.

appear to be highly faceted to a specific crystal lattice direction. After electron beam irradiation (Fig. 4b), the edges are etched to form sharp zigzag-faceted terminations (Fig. 4c). The zigzag edges have iodide atoms protruding out on one side and lead atoms on the adjacent edge of the same flake (Fig. 4d, e). Metal and chalcogenide-terminated zigzag edges have been routinely observed for as grown monolayer TMDs, such as MoS₂[55,56]. However, in this case, the edge etching also happened in directions which maintained the zigzag-PbI₂ alignment to the armchair graphene direction (Supplementary Note 12).

Edge etching occurred by atomic loss under the electron beam (Fig. 4h–i), and edges also showed atomic attachment to restructure (Fig. 4j–l, Supplementary Note 13). After 30 s of electron beam exposure, the atoms at the edge are sputtered out, by unzipping the outer most atoms to retain its uniform zigzag termination (Fig. 4i)[56]. Prolonged exposure to the electron beam eventually leads to edge roughening. DFT and ab initio molecular dynamics simulations have shown the edge

reconstructions effect the electrical and magnetic properties of the material[57].

ADF-STEM of the central lattice region of the PbI₂ monolayers revealed point defects across the region (Fig. 5a). Single-point vacancies were observed for both the lead and iodide sub-lattice sites in PbI₂ (Fig. 5b, c), and identified using line profiles (Supplementary Note 14). The energy to remove a single iodide or lead atom from the lattice into the vacuum was calculated by DFT to be 3.15 and 6.36 eV, respectively (Supplementary Note 15). The maximum amount of energy ($E$) transferred by an incident electron (@80 keV) to a single iodide or lead atom is calculated to be 1.38 and 0.85 eV, respectively[58]. This energy transfer from the incident electron to the specimen atoms is much lower than the required sputtering energy calculated by DFT. Hence, we can deduce that the mechanism of damage in this system is more complicated than purely knock-on damage from atomic sputtering, and likely involves radiolysis. This is not surprising, since PbI₂ is a large bandgap semiconductor and

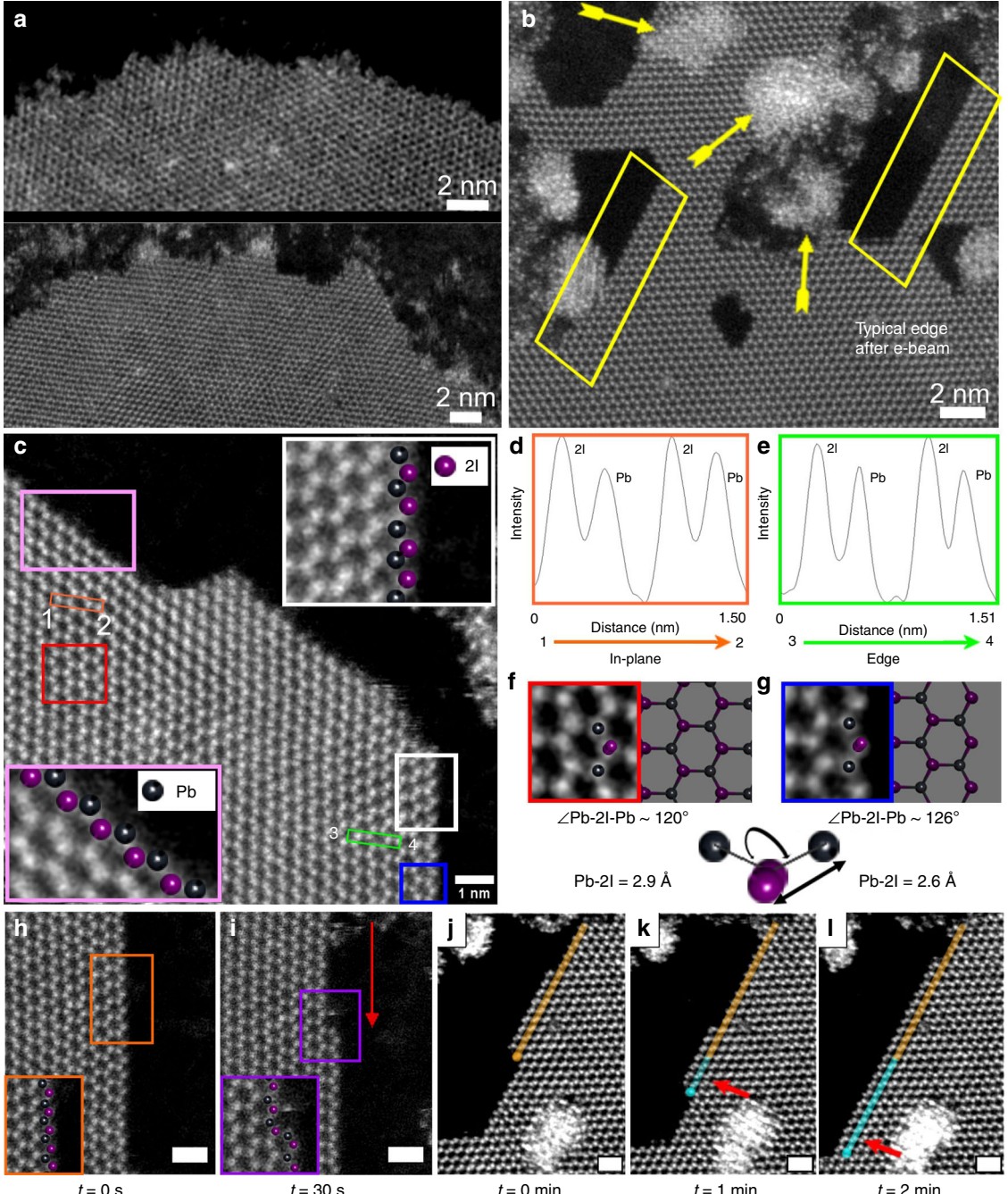

**Fig. 4 Edge study of PbI₂.** ADF-STEM image of **a** typical intrinsic edges of two different monolayer PbI₂ flakes just after deposition on the graphene and **b** sharp zigzag-faceted edges of the monolayer PbI₂ flake after electron beam irradiation. The arrows show the clustering of the atoms after having been sputtered out of the edges. The yellow boxes show the faceted zigzag edge formation from the intrinsic edges after the damage from electron beam. **c** ADF-STEM images showing I- and Pb-terminated zigzag termination. The two insets in the image shown with white and pink boxes show the high resolution of the areas with same color annotation that have I- and Pb-terminated edges. The line profile of the regions marked in orange and green boxes confirms the position of the lead and iodide in the honeycomb structure. **d, e** The line profile of the orange and green boxes shown in **c**, in the direction of 1–2 and 3–4. **f** High-resolution ADF-STEM image of the area shown under red box in **c**. **g** High-resolution ADF-STEM image of the area shown under blue box in **c**. **h, i** Time-lapse ADF-STEM image of the edge at $t = 0$ s and $t = 30$ s of electron beam exposure. The red arrow indicates the ejection of the electrons from the edges due to the damage from the electron beam, leading to 'unzipping' of the chain, maintaining the zigzag edge of the flakes. **j–l** Time-lapse series of ADF-STEM images recorded after 1 min of electron beam exposure. The yellow annotations show the edge-terminated bonds at the edge at $t = 0$ s, which starts reconstructing in **k**, annotated with blue. After 2 min, (**l**) the edge was fully reconstructed, as shown with growing blue annotated region. The scale bars in **h–l** correspond to 1 nm.

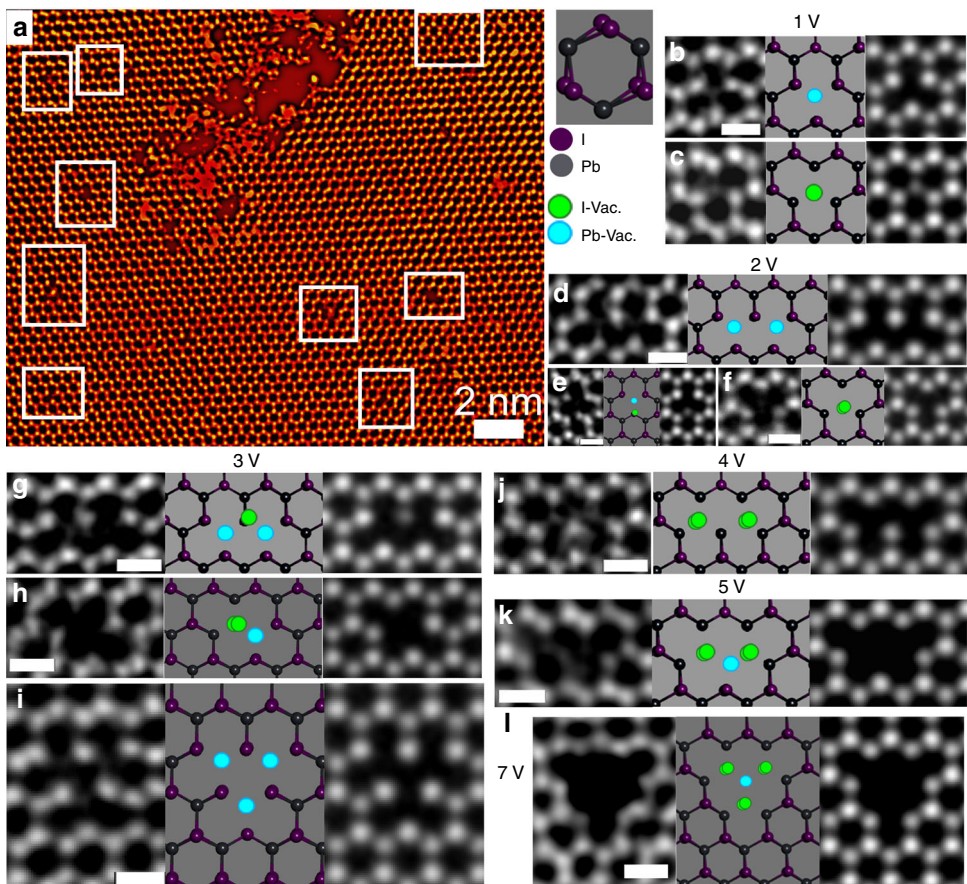

**Fig. 5 Vacancy study in monolayer PbI₂. a** Low-magnification ADF-STEM image of monolayer PbI$_2$ region after electron beam exposure. The purple and gray spheres correspond to lead and iodide in the structure. The green and blue spheres correspond to the iodide and lead vacancies in the system. ADF-STEM images and their corresponding 1 H PbI$_2$ structured atomic model, as well as multislice simulations have been shown for **b** point defect with a missing lead atom, **c** point defect with a missing iodide atom, **d** double-point defect with two iodide atoms missing, **e** double-point defect where two lead atoms from adjacent crystal cells are lost, and **f** double-point defect with one lead and one iodide atoms missing. It can be observed that the single iodide has lower intensity than that of lead or two iodides, **g** three vacancies formed out of two lead and one iodide loss. It can be observed that the remaining single iodide in the middle has lower intensity than that of lead or two iodides, **h** three vacancy formed by the absence of three lead atoms from adjacent crystal cells, **i** another three vacancy formed by two iodide and one lead loss, **j** four vacancy defect formed by loss of two iodine from adjacent honeycomb structures, **k** five vacancies formed because of loss of four iodide atoms and a lead atom in a row, **l** bigger triangular defect formed because of the loss of one lead and six iodide atoms from the center. The scale bar in all the images indicate 0.5 nm.

therefore prone to ionization effects from the electron beam. Several different types of multiple atom point vacancies are identified, ranging from two atoms (Fig, 5d, e) to the seven atom defect clusters (Fig. 5k). Vacancies did not appear to assembly into linear structures, as in TMDs, but instead aggregated to form clusters (Supplementary Note 16).

We also observed multiple occasions where the defect (Fig. 6) had one Pb atom in the center and five iodide atoms surrounding it, which was stable even under prolonged electron beam irradiation (Fig. 6a–d). Figure 6f, g,i, j shows the proposed atomic model and the simulation of the two different configurations of this defect, (Fig. 6e, h) respectively. Since a single iodide atom is much lighter ($Z = 53$) than the single Pb atom ($Z = 82$), it has lower contrast in ADF-STEM. The line profiles in Fig. 6 confirm the presence of iodide atoms close to the lead atom in the center, causing the iodide peaks to be only partially resolved and forming a tail either side of the lead atom. Figure 6x shows the structure of the system, where Pb is coordinated with five other iodide atoms with inter-atomic distance varying from 1.4–2.3 Å. We observed the formation of a stable four-membered ring around the lead atoms (Fig. 6f, i), where the lighter iodide atoms

substitute the heavier lead atoms (Fig. 6f). Similar stable four-membered ring configuration have also been reported for monolayer MoS$_2$ structures[59].

During electron beam irradiation, the vacancies migrated through the lattice and reconstructed. We performed DFT calculations for vacancy migration and found that the energy barrier for migration of lead atom, the in-plane migration of the iodide atom and the out-of-plane migration of iodide atom that corresponded to 0.32, 0.22, and 0.07 eV, respectively (Supplementary Note 17). The energy barrier for migration of the atoms on the surface is considerably lower than the energy imparted from the electron beam and thus explains the quick migration of the vacancies and defects in the system. Figure 7 is an example of a double iodide vacancy (Fig. 7b) and its migration (Fig. 7c) before subsequently healing to render the original structure in Fig. 7a, d. Defect healing occurred in point defects and also with in multiple atom vacancies, such as Fig. 7e–g, where a large defect with seven atoms missing (one Pb and six I) is healed.

Further electron beam exposure caused the vacancies grow larger and form nanopores in monolayer PbI$_2$ crystals[60,61]. The nanopores undergo geometric reconstructions by atomic loss

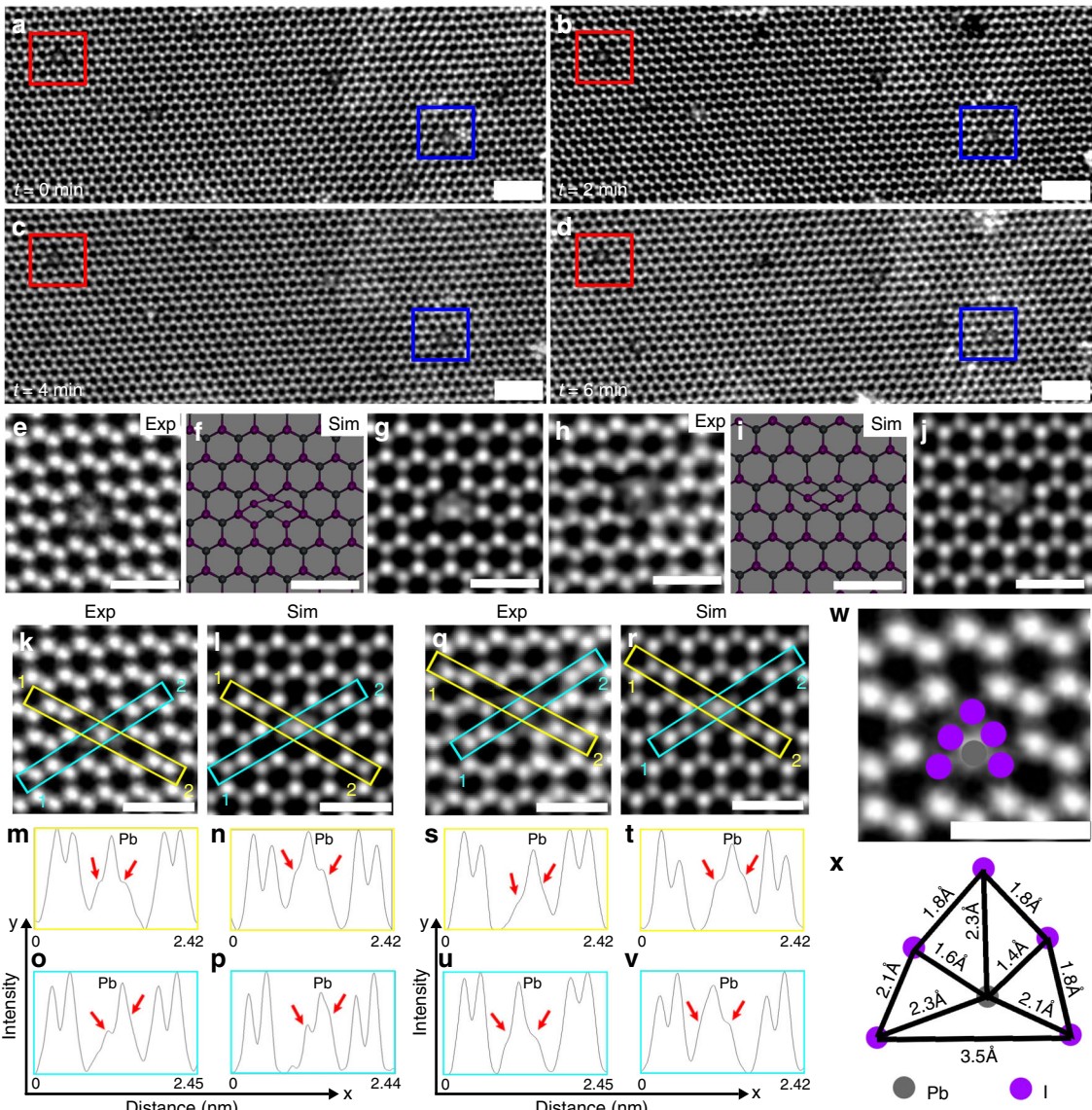

**Fig. 6 Stable defect dynamics in PbI₂. a–d** Time-lapse ADF-STEM images taken after every 2 min. The red and blue boxes show the two different orientations of the substitutional defect, which is rather stable under the electron beam exposure. **e** ADF-STEM image of a defect with lead in the center and surrounded by five other iodide atoms. **f** Atomic model that shows the atomic model of the corresponding experimental result. **g** Multislice image simulations based on the atomic model in **f**, schematically illustrating the lower intensity of the single iodide atoms surrounding the lead, which agrees with the experimental data in **e**. **m, n** Line profile taken across the lead and iodide atoms across the yellow annotated regions in the experimental data **k** and multislice simulated image in **l**. The red arrow shows unresolved iodide peaks around the lead. **o, p** Line profile taken across the lead and iodide atoms across the blue annotated region in **k** and **l**, respectively. The red arrow shows unresolved iodide peaks around the lead. **j, h** ADF-STEM image of the opposite orientated substitutional defect. **i** Atomic model that shows the atomic model of the corresponding experimental result. **j** Multislice image simulations based on the atomic model in **i**, schematically illustrating the lower intensity of the single iodide atoms surrounding the lead, which agrees with the experimental data in **h**. **s, t** Line profile taken across the lead and iodide atoms across the yellow annotated regions in the experimental data **q** and multislice simulated image in **r**. The red arrow shows unresolved iodide peaks around the lead. **u, v** Line profile taken across the lead and iodide atoms across the blue annotated region in **q** and **r**, respectively. The red arrow shows unresolved iodide peaks around the lead. **w** Annotated image of **e**. **x** Distance between the different atoms shown in the annotated region in **w**. Scale bars in **a–d** correspond to 2 nm, whereas in the rest of the images correspond to 1 nm.

(Fig. 8a–p), but also sometimes gain atoms (Fig. 8o–p). Diffusing Pb and I atoms add to the edge sites of the nanopores to enable reconstruction. The nanopores in general adopt zigzag edge terminations and have triangular or hexagonal shape. In several cases, atoms migrate from one side of the nanopore to another side to change the nanopore shape. Figure 8q–u shows a case, where a large triangular nanopore is formed by electron beam

irradiation, and is then backfilled by its nearby atoms. Between Fig. 8s, t, the top left section of the triangular nanopore has contracted by atoms filling it in from the nearby edge region. The red arrows indicate the possibility of migration of the atoms from one position to the other. The electron beam damages the material and forms the nanopore but at the same time, we are able to see the fast migration of atoms and the self-healing

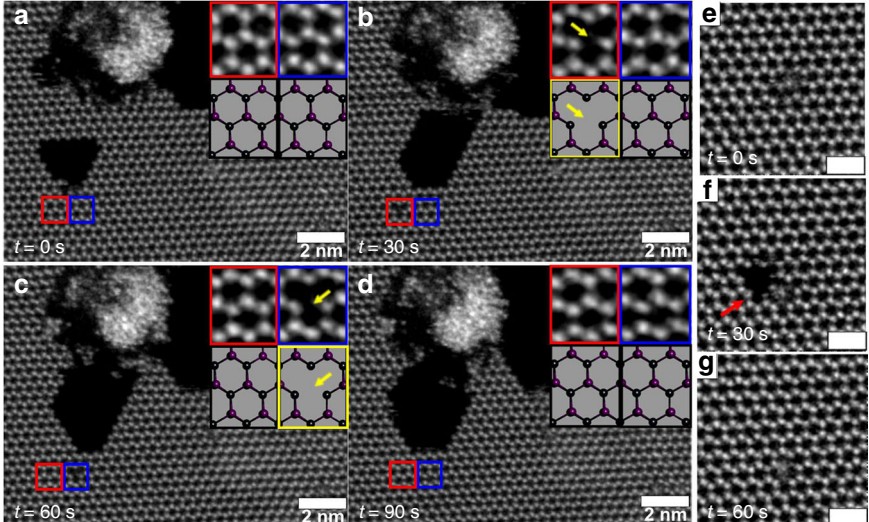

**Fig. 7 Point defect formation, migration, and then reconstruction. a–d** Time series of ADF-STEM images recorded after every 30 s, of point defect formation, migration, and then reconstruction. The inset under red and blue boxes in the images are high-magnification images of the defects that have been marked with smaller red and blue boxes, respectively, in the same image. The inset figures in black and yellow boxes below the high-magnification insets are the corresponding atomic models. The yellow boxes and arrows in **b** and **c** highlights the formation of a point vacancy and its migration in the inset above, before finally reconstructing in **d**. **e–g** Time series of ADF-STEM images recorded at an accelerating voltage of 80 kV after every 30 s of electron beam exposure. The red arrow in **f** shows the formation of seven vacancy defect, but it gets quickly reconstructed in image **g** after another 30 s of exposure. The scale bars in **e–g** correspond to 1 nm.

process in the material in real time. However, prolonged exposure to the electron beam ultimately leads excessive sputtering of atoms than the self-healing itself leading to a bigger nanopore formation. Similarly, in Fig. 8q–u, the self-healing is not perfect, but the nanopore is no longer present and vacancies have been eliminated by the migration to the edge.

We also observed the migration of entire nanopores as a single entity, up to 4 nm on the same monolayer flake. Figure 9a–i is set of time lapsed images, taken every 30 s, of hole formation and reconstruction as well as migration on the same monolayer flake upon beam exposure. The yellow boxes in Fig. 9g–i show the accumulation of the atoms at the edge and surface region next to the nanopore. The quick variation in the position and size of these holes clearly show how mobile the lead and iodide atoms are on the surface and continuously try to form new structures.

## Discussion

We have successfully shown the liquid exfoliation of $PbI_2$ using chloroform and formation of >99% monolayer nanodisks. This approach has the potential to be scaled up to produce a large amount of exfoliated $PbI_2$. Direct imaging of the monolayer $PbI_2$ nanodisks is challenging using conventional TEM grids, and the use of graphene was critical for high-resolution ADF-STEM imaging. However, the van der Waals interactions with the underlying graphene lattice causes epitaxial alignment of the $PbI_2$ and leads to the observation of the 1 H crystal phase for $PbI_2$, with no observation of 1 T phases on graphene. The relative lattice spacing of $PbI_2$ and graphene results in preferential alignment of the $PbI_2$ zigzag direction to the armchair direction of graphene to maximize the commensuration of the two lattices. Integrating monolayer $PbI_2$ with other 2D materials provides a huge scope for many applications and the ability to manipulate its atomic monolayer structure provides a unique opportunity to tune its properties. Studying the influence of the substrate on the structure of $PbI_2$ would hopefully make it easier to understand the complicated polytypes and stacking arrangement in $PbI_2$.

ADF-STEM imaging of the $PbI_2$ monolayers revealed highly mobile point vacancies and their growth into nanopores, which exhibited reconstructions and self-healing. Studying these vacancies in the 2D material is important for the foundation knowledge of the semiconducting properties for future work.

## Methods

**Chemical vapor deposition growth of graphene.** Graphene was synthesized by CVD on a copper substrate. A piece of copper foil was first sonicated in a diluted hydrochloric acid solution (1 mol/L) to remove the oxides on the copper surface. The copper foil was then sonicated in deionized water, acetone, and isopropanol for 5 min each to remove organic residues. To ensure uniformity of temperature during the whole growth process, the copper foil was positioned in the center of a furnace. A total of 1% methane in argon ($CH_4$), 25% hydrogen in argon ($H_2$), and 100% argon (Ar) were used for graphene synthesis. Before the reaction started, the system was first purged with 1000 sccm Ar, 500 sccm $H_2$, and 100 sccm $CH_4$ for 30 min. The furnace was then heated to 1060 °C with a ramp rate of 50 °C/min accompanied by a flow of 500 sccm Ar and 100 sccm $H_2$. When the temperature reached 1060 °C, the copper was then annealed with the same flow rate for 1 h. Following the annealing process, the synthesis was carried out at 1060 °C for 1 hr with a flow of 500 sccm Ar, 100 sccm $H_2$, and 5 sccm $CH_4$. The furnace was removed from the sample, which allowed the sample to be fast cooled to room temperature.

**Liquid-phase exfoliation.** $PbI_2$ powder from Sigma–Aldrich (99% purity) was added to chloroform to form a solution of concentration 1 mg/ml (~0.2 mM). The solution was sonicated in an unheated ultrasonic water bath (Ultrasonics effective output of 200 W; Ultrasonics peak power of 600 W; Ultrasonics operating frequency 30–40kHz) for ~30 min to make sure that all the lead iodide had exfoliated and dispersed in the solution. The dispersion was then left to stand for 24 h to let the bulk particles to settle down at the bottom of the container. Then, the supernatant of the dispersion (from the top) was extracted with a pipette and used for preparing TEM samples.

**TEM sample preparation.** Graphene was transferred onto a lacey carbon copper 400 mesh TEM grid using a poly (methyl methacrylate (PMMA) support and then the PMMA removed by acetone. It was then subsequently cleaned by Ar/$H_2$ and vacuum annealing. A few drops (10–20 µl each) were taken from the supernatant of the dispersed solution of 1 mg/ml concentration of $PbI_2$ in $CHCl_3$ and dropcasted on to the TEM grid containing suspended graphene. The TEM holder was then baked in high vacuum at 60 °C to remove all the solvents and other impurities from the surface.

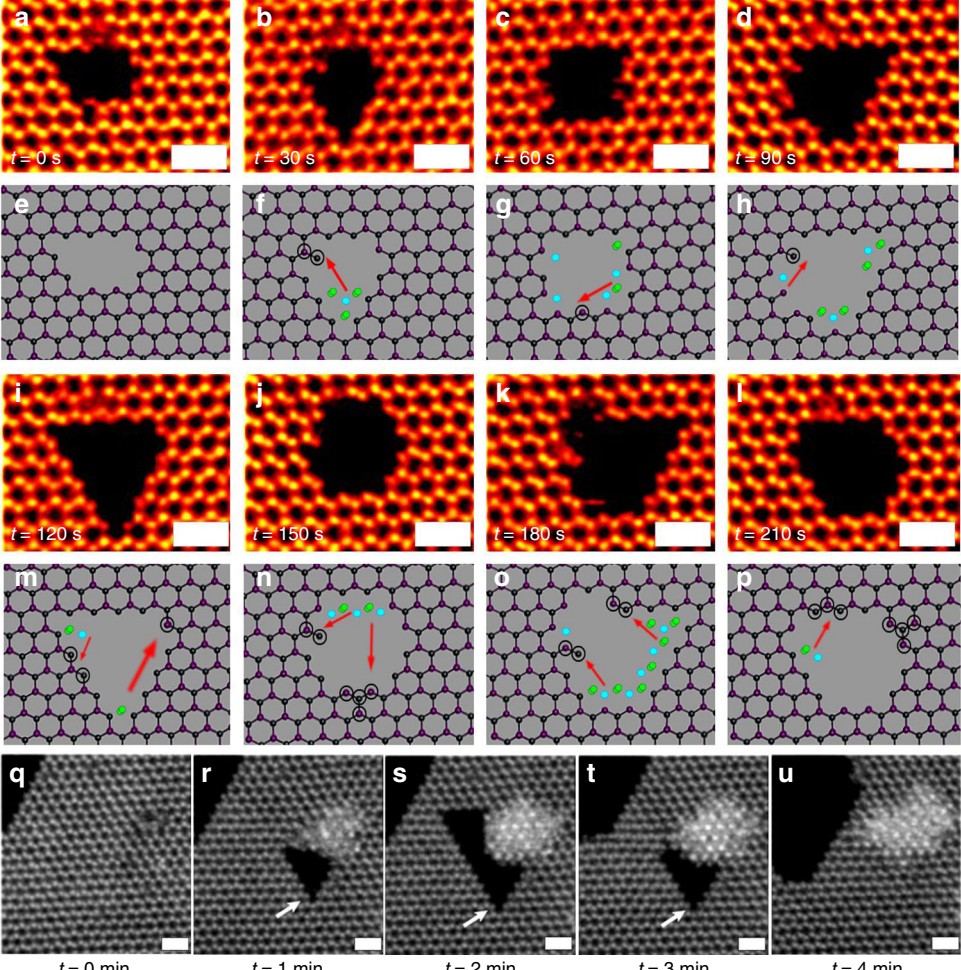

**Fig. 8 Nanopore geometric reconstructions. a–p** Time series of ADF-STEM images, and their corresponding atomic models. Holes can be seen to change its shape in monolayer PbI$_2$ flake. The blue atoms correspond the lead defect region and the green corresponds to the iodide defect region. The black circles around the atoms in the atomic models correspond to the new atomic bond in the structure as compared from the previous frame. **q–u** Time series of ADF-STEM images recorded after every 1 min of electron beam exposure. The defect starts forming after 1 min, in image **r** and more atoms are ejected to form a bigger defect in **s**. Then the atoms at the edge starts ejecting out in **t** and reconstruction of the flake starts occurring leading to again defect-free surface in **u**. The scale bar in all images corresponds to 1 nm.

**Scanning transmission electron microscopy**. Room temperature ADF-STEM imaging was performed using a JEOL ARM200F located at the David Cockayne Centre for Electron Microscopy (DCCEM) within the Department of Materials at the University of Oxford. An acceleration voltage of 80 kV was used for taking all the images, unless stated otherwise. Imaging conditions used a 30 μm condenser lens (CL) aperture with a convergence semiangle of 22.5 mrad and a beam current of 35 pA. The acquisition angles for these images was 70–220 mrad.

**Data processing**. Images were processed using the ImageJ software. Atomic models were constructed using Accelrys Discovery Studio Visualizer. Multislice image simulations were performed using JEMS. PbI$_2$ crystal structure was drawn using Crystal Maker Version 10.3.1.

**DFT calculation**. The structural, energetic, and defect migration calculations in this work are based on DFT implemented in the Vienna Ab-initio Simulation Package (VASP)[62,63]. A $10 \times 10 \times 1$ supercell is used for calculating the sputtering energy and defect migration barriers, and two more different systems are designed to probe the short-range angle dependence of stacking energy: a hexagonal PbI$_2$ flake (side length four PbI$_2$ unit cells, 1.8 nm) on continuous graphene substrate. The systems are varied with different angles between PbI$_2$ and graphene (i.e., armchair to zigzag stacking order). The elements are represented by projector-augmented wave potentials with 500 eV energy cutoff and the four ($6s^26p^2$) valence electrons for Pb, seven ($5s^25p^5$) for I, and four ($2s^22p^2$) for C are treated explicitly[64,65]. Initial relaxation and energetics are calculated via the generalized gradient approximation, Perdew–Burke–Ernzerhof, augmented by DFT-D3 method with Becke–Johnson

damping for the interlayer dispersive interaction[66,67]. The first Brillouin zone is sampled by the tetrahedron method on a $4 \times 4 \times 1$ $k$-mesh for the $10 \times 10 \times 1$ supercell, and only gamma point for the angle-dependence studies. All structures are relaxed until the force on each atom is <0.01 meV/Å. The sputtering energy is defined by the energy difference between the crystalline structure and the one with a vacancy, for instance, the I sputtering energy is $\Delta E_I = E_{crystal} - (E_{V_I} + E_I)$. The energy tolerance is set to 0.0001 for all energy calculations. The minimum energy paths for defect migration are determined by the nudged elastic band (NEB) method[68,69]. A total of $10 \times 10$ supercells and $4 \times 4$ $k$-point grids are employed, and a criterion of 0.03 eV/Å is set for the force convergence. For $V_I$ migration, two pathways are considered: in-plane and out-of-plane directions, and only in-plane $V_{Pb}$ migration is considered. The starting and end points for NEB search are prepared by removing the corresponding atoms from the relaxed pristine structure, and then further relaxed. The six intermediate images are interpolated by the Vasp TST toolkit[70]. The converged energy paths are the fitted by cubic spline functions.

**Supporting information**. Effect of sonication duration and dropcasting, monolayer and bilayer PbI$_2$ intensity contrast, graphene sample without PbI$_2$, different sizes of monolayer PbI$_2$, proof of monolayer flakes instead of bilayer AA′ 1 H structure, T-phase structure on lacey carbon TEM grid, ADF-STEM images of structural transformation immediately after dropcasting on graphene, Moire pattern of 1 T and 1 H atomic models on graphene, zigzag alignment of PbI$_2$ with graphene after edge etching, relative alignment of time frame of PbI$_2$ etching from Fig. 4, line profile from Fig. 5, DFT calculation of sputtering energy of lead and iodide, accumulation of lead particles after etching, and DFT calculation of vacancy migration.

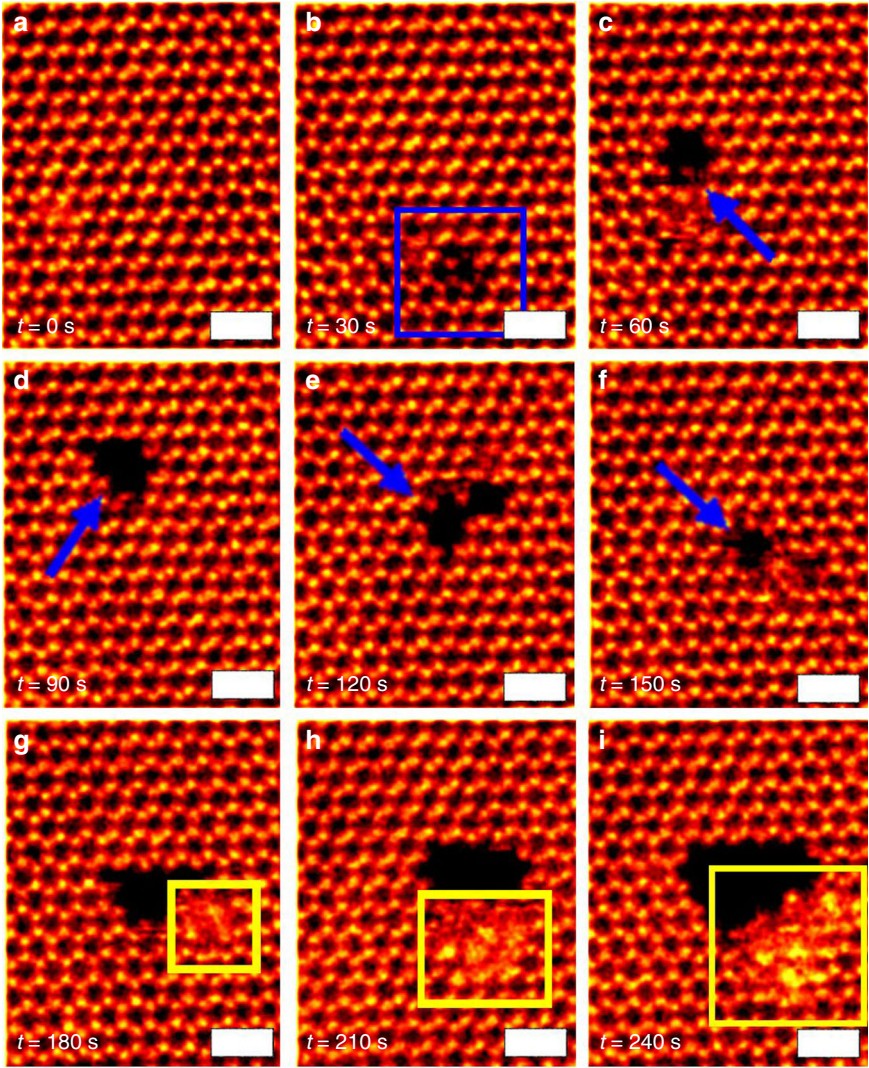

**Fig. 9 Nanopore migration. a–i** Time series of ADF-STEM images recorded of nanopores changing form, as well as migrating large distances within single-monolayer $PbI_2$ flake. The scale bar corresponds to 1 nm.

## Data availability

Experimental procedures and characterization data are available within this article and its Supplementary Information file. The data that support the findings of this study are available from the corresponding authors upon reasonable request.

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

## Acknowledgements

J.H.W. thanks the support from an ERC Consolidator Grant (LATO 725258). S.S. thanks the joint committee of Linacre College and MPLS division of the University of Oxford for financial support through the Women in Science Scholarship and EPSRC studentship. The authors acknowledge use of characterization facilities within the David Cockayne Centre for Electron Microscopy, Department of Materials, University of Oxford.

## Author contributions

S.S and J.H.W conceived the idea for the project, designed the experiments and analysis procedure. S.S. carried out all the chemical synthesis and experiments. S.S. carried out analysis of data and wrote the manuscript with contributions from J.H.W. Y.S. synthesized monolayer graphene. T.Z. and A.F.L carried out the DFT calculation. J.C.G. and K.P. provided materials for the experiments and helped with scientific discussions. J.H.W. supervised the project. All authors discussed the results and commented on the manuscript.

## Competing interests

The authors declare no competing interests.
