## [Peer Review File · Nature Communications]

Reviewers' comments:

Reviewer #1 (Remarks to the Author):

The paper by Sinha et al. discusses synthesis of 2D monolayer Lead Iodide on graphene substrate using liquid exfoliation method, and further explored the atomic structure, defect structures and structural change under electron beam using scanning transmission electron microscopy. Despite that the authors put lots of interesting information in one paper, the paper still lacks the depth and novelty that are required by a journal like nature communications. The results are 'routine' and there is very loose relationship between different sections. The defect structure, edge structure, electron beam effect and self-healing are all well-known effect for 2D materials under the electron beam. For example, under electron beam irradiation, holes can form for basically any 2D materials. The hole expansion process is more or less the same for all the TMDs. The edge structure is mainly zigzag for TMDs, which is not surprising that such edge structure is also observed for monolayer PbI₂. The atoms are extremely mobile around defects and could form interesting transient structures, leading to self-healing or hole expansion depending on the local structure. The paper reports lots of observations, but I am not sure how those observations are special for PbI₂, and how those observations extend our current understanding of defect structure of 2D materials, under electron beam. And it is not clear how the observations can be related to the physical or functional properties of PbI₂, as the authors have nicely discussed in the introduction, such as band gap, mobility. To summarize it, this is a decent characterization paper for monolayer PbI₂ synthesized using liquid exfoliation, but it does not really provide more information than we have already known about this 2D material. This paper would fit better in ACS Nano.

Reviewer #2 (Remarks to the Author):

This manuscript describes the preparation and structural analysis, via electron microscopy, of the 2D transition metal dichalcogenide system PbI₂, a candidate material for optoelectronic applications in the green-to-UV region of the spectrum. The 2D form of this material has received less attention than its multilayered form, in part due to the difficulties of obtaining high quality monolayers. Main results include the preparation of the monolayer (2D) form in a manner amenable to electron microscopy imaging, its adoption of a preferred phase and orientation relation with the graphene substrate, and the observation of various beam induced defect structures and their evolution over time. I will leave any criticisms warranted of the materials preparation to reviewers better qualified to comment on it, and focus on the electron microscopy.

Though adequate signal-to-noise is a challenge for any thin sample, as evident in the images shown, PbI₂ is well-suited to annular dark field scanning transmission electron microscopy (ADF-STEM) because the strong scaling of the signal with atomic number allows ready discrimination between the Pb and I atoms (with low contribution from the light graphene support), and the structure of the two phases (T-phase and H-phase) are clearly distinguishable by ADF-STEM contrast. Drawing not only on the directly interpretable nature of ADF-STEM for thin samples but also on comparison with electron scattering (multislice) simulations, the authors demonstrate the phase and monolayer nature of their prepared samples and proceed to catalogue a range of (beam-induced) structural defects, many of which evolve dynamically under electron irradiation but some of which, once formed, seem reasonably stable. The figures are generally of high quality and used to good explanatory effect. While this kind of electron microscopy analysis is no longer unusual, the present analysis seems to me generally done well, and the significance of the material should help this work appeal to a wide cross-section of the materials research community. I would ask the authors to consider the points raised below by way of minor revision before the paper be accepted for publication.

Main concern:

Though some motivation is provided in the introduction and conclusion, the discussion of Figs. 4–9 provides a great enumeration of defect structures encountered, but does not convey to me a sense of conceptual framework or practical significance. Especially given that these defects seem to be electron-beam induced, I feel the manuscript would be significantly strengthened if the unity of purpose and significance of these findings could be more clearly conveyed.

Minor considerations:

* Line 132: given the time between images shown is many orders of magnitude greater than most atomic dynamics, that holes are seen in Fig. S3 (rather than, say, “thinning”) seems to me to be a less than conclusive demonstrating that the sample is indeed only a monolayer thick. Could the authors please elaborate on why holes would be expected to open up layer-wise?

* Line 168–170: “The smaller the difference in the relative lattice spacings between the two crystals, the larger the van der Waals interaction is likely to be. The best lattice match occurs when the PbI₂ adopts 1H phase and is aligned to the arm-chair direction, which agrees with our experimental findings.” Please consider spelling out this connection in greater detail in reference to Fig. 3(s-v), especially since the green lines indicating the “minimum distance where Pb atom overlaps a C-C bond in graphene, indicating loss of commensuration” seem to me to be of comparable length between Figs. 3(s) and (v).

* In Fig. 4(j-l), what was used as the basis of relative alignment? Given the mobility of defects asserted elsewhere, it is not entirely obvious to me that the orange lines on those three successive images correspond to the same set of atoms.

* The weak contrast at the centre of the defect in Fig. 5(b) is interpreted to indicate a single vacancy (rather than two I vacancies one above the other). However, there seems to me to be similar remnant contrast present in the centre in Fig. 5(b), (j) and (k) but those cases are interpreted as complete vacancies. Please elaborate on how the interpretation / assignment of these experimental images to particular simulated configurations was made.

* In Figs. 8(f-h), (n-p), what is the significance of the red arrows? It seems to imply a direct this-atom-moved-from-here-to-here interpretation that I would consider questionable given the time frame of atomic dynamics relative to the time elapsed between images.

* In the supplementary material, I find the discussion comparing what Fig. S4 shows to what Fig. S5 shows confusing, especially given the differences in format of the two figures. Please clarify.

* In section S5, I find it hard to make the connections between Figs. S5(c) and (d) on the basis of relative column intensities, though the intercolumn spacing is persuasive. Should this be explicitly stated?

* I also find it hard to see Fig. S6(c) as being the “high resolution image of the area boxed in red color in figure a” – not least, the extent of the fields of view appear to be quite different. Please confirm the correct image has been provided.

Trivial points:

* In line 222, “The red arrow indicates the ejection of the electrons...” should presumably refer to the ejection of atoms? (At the risk of being especially pedantic, I’d also argue that what is shown in Fig. 4(i) is a red line, not an arrow since it lacks a head.)

* In line 380: "Sigma Aldrick" should read "Sigma Aldrich".

* In the supplementary material, there are currently two figures both labelled Fig. S5.

Reviewer #3 (Remarks to the Author):

The authors for the first time reported the synthesis of 2D monolayer Lead Iodide nanodisk with diameter of 30-40 nm using liquid-exfoliation method. Then interface it with graphene to form van der Waals heterostructure which may be important for future applications in opto-electronics. Defective dynamic in 2D PbI₂ is also studied. The topic is interesting. All the experimental synthesis and characterization are basically sound. I would recommend it publishing in Nature Communications after some minor revisions.

(i) The band gap of PbI₂ is quite large. The Most practical applications should be discussed.

(ii) Many defects have been formed and observed. Which has the lowest formation energy?

Reviewer #4 (Remarks to the Author):

In this work, Jamie H. Warner and co-workers report the liquid-exfoliation of PbI₂ nanodisks and systemically study the interaction between PbI₂ and graphene. The fundamental atomic structure and different defects as well as the edge states under electron beam irradiation are well studied. The method is simple, and the solid experiment has been conducted by ADF-STEM. The interaction between graphene and PbI₂ is illustrated. Some comments, especially the detailed structure information, should be addressed before its acceptance.

1) The authors claim that only 2 flakes with bilayer structure are found among more than 300 flakes. However, in figure 1c, 1e and figure s1 and s2, there are small flakes showing thickness is above bilayer. The authors use the supernatant of the dispersion for preparing TEM samples. Where (bottom or top) is the dispersion taken from the container? Do the flakes have the similar thickness in the bottom or top of the container? The authors should consider all the few-layer PbI₂ not only the monolayer and bilayer. Meanwhile, the experiment conditions of liquid-exfoliation for the 300 flakes should be provided in the manuscript.

2) Why the PbI₂ prefers the orientation of zigzag aligned to graphene armchair direction rather than the graphene zigzag direction. The detailed reason should be discussed and offered in the manuscript.

3) The authors claim that the best lattice matching occurs in the 1H-PbI₂ with direction aligning to the graphene arm-chair. On the contrary, what's the structure of PbI₂ (1H or 1T) when its direction aligning to the graphene zigzag?

4) What's the graphene direction when the edges of PbI₂ are etched to form sharp zig-zag faceted terminations after electron beam irradiation?

5) Different defects are induced by electron beam irradiation such as 4-membered ring configuration, atoms missing. As for the defects, the significant issue is the stability. Do these structures can be stable in air?

6) What's the effect of the accumulation of the atoms in Fig (i-j). To form a new material or only a new structure of PbI₂? It should be clarified.

Reviewer#1

The paper by Sinha et al. discusses synthesis of 2D monolayer Lead Iodide on graphene substrate using liquid exfoliation method, and further explored the atomic structure, defect structures and structural change under electron beam using scanning transmission electron microscopy.

Reviewer#1 1) Despite that the authors put lots of interesting information in one paper, the paper still lacks the depth and novelty that are required by a journal like nature communications. The results are 'routine' and there is very loose relationship between different sections. The defect structure, edge structure, electron beam effect and self-healing are all well-known effect for 2D materials under the electron beam. For example, under electron beam irradiation, holes can form for basically any 2D materials. The hole-expansion process is more or less the same for all the TMDs.

Our response: We have now revised our manuscript to emphasis the novelty and make clear the depth of our findings. It has been now structured in three sub-parts which describes the (a) synthesis of predominantly monolayer PbI_2 (b) 1-H atomic structural phase of PbI_2 (c) atomic scale defects. All these results are novel and provide insight into the special structure of PbI_2 . The results are new as well as quite different from other 2D materials. Hole expansion process is not same for PbI_2 , as we have seen a big restructuring, very stable new defects as well as different pathway of hole formation. Different hole expansion process can lead to different types of defect formation on large scale and different influence on the semiconducting properties, and thus makes all the 2D materials different on the monolayer scale. We have also replied to this in more details with the 3rd question. We have now restructured our paper as well as added more comparisons with other 2D material in our main text to strengthen the paper.

Reviewer#1 2) The edge structure is mainly zigzag for TMDs, which is not surprising that such edge structure is also observed for monolayer PbI_2 .

Our response: PbI_2 is not a TMD, so it is not valid to assume that the behaviour seen in TMDs will apply to PbI_2 . For example, in other 2D materials such as Graphene, a mixture of arm-chair and zig-zag edges can exist. To date, there is no experimental evidence on the nature of edges in PbI_2 monolayers, so prior to our report, it is only theoretical predictions. Our results provide the very first evidence on the edge terminations on PbI_2 monolayer crystals. In our edge-study the PbI_2 maintains the zigzag structure to the armchair direction of the graphene even whilst etching. It etches away in three possible directions which is specific to graphene orientation. This is very unique and we have added more information to the main text as well as added a new figure in S.I. 11.

Reviewer#1 3) The atoms are extremely mobile around defects and could form interesting transient structures, leading to self-healing or hole-expansion depending on the local structure. The paper reports lots of observations, but I am not sure how those observations are special for PbI_2 , and how those observations extend our current understanding of defect structure of 2D materials, under electron beam.

Our response: We now include new DFT calculations of the sputtering energy for Pb and I atoms in monolayer PbI_2 and compare these to the energy transferred by the electron beam. We also now include DFT calculations of the vacancy migration barrier for Pb and I vacancies in PbI_2 . These new results show that energy from the electron beam is not sufficient to sputter the Pb or I atoms, indicating that more complex vacancy mechanisms are occurring than knock-on damage alone. This includes ionization effects due to the large band gap of PbI_2 , known as radiolysis. We do show that the energy from the electron beam is more than sufficient to drive the vacancy migration. We observe both Pb and I vacancies, which are both mobile and this is

different to most other 2D crystals made up of one heavy and one lighter element. This leads to unique vacancy and hole formation in PbI_2 with rapid vacancy migration.

Furthermore, PbI_2 is of special interest, because 3D crystal structures of PbI_2 (2H and 4H) have been shown to comprise of 15-20% atomic positions and have been shown to change the structure.^{1,2} This can just give an idea of how much importance the vacancies and defects are, in 2D- PbI_2 and can have important consequence for its semiconducting properties. On this front, our work is novel, since no one has studied and characterized the atomic scale defects and vacancies of PbI_2 at the atomic scale. In addition, it exhibits new kind of stable defects (figure 6) which is unlike any other 2D materials and may provide new insights into the bonding properties of lead and iodide.

Importantly, the hole formation differs from other 2D materials studied at room temperature. Defects and hole formation and propagation for different 2D materials have shown vastly different behaviour. For instance, monolayer hBN has nanopores that form and grow in size whilst maintaining triangular shape from a monovacancy.³ Graphene has nanopores at room temperature that are generally round. At room temperature in TMDs, such as MoS_2 and WS_2 , round holes form that are not well faceted shapes and the heavy metal aggregates around the hole edges and can form nanowires. However, in PbI_2 , the holes form well faceted shapes with zig-zag termination at room temperature, and this is due to the high mobility of vacancies under the electron beam and the electron beam induced displacement of both Pb and I atoms.

These results provide a detailed insight into the atomic structure and defects in monolayer PbI_2 . We have now added this to the main text to make the paper more informative.

Reviewer#1 4) And it is not clear how the observations can be related to the physical or functional properties of PbI_2 , as the authors have nicely discussed in the introduction, such as

band gap, mobility. To summarize it, this is a decent characterization paper for monolayer PbI₂ synthesized using liquid exfoliation, but it does not really provide more information than we have already known about this 2D material. This paper would fit better in ACS Nano.

Our response: Knowing the defects and the structure of the defects of 2D materials is crucial to understanding its effects on its physical and chemical properties. The accurate determination of the atomic structures of these defects and vacancies can contribute to precise estimation of its effects on the transport properties of the material. The impact of the strong van der Waals interaction with graphene is yet another important part of the work and is crucial while considering it for future applications in opto-electronics.

References:

- (1) The Structure of PbI₂ Polytypes 2H and 4H: A Study of the 2H-4H Transition. *J. Phys. Condens. Matter* **1990**, 2 (24), 5285–5295.
- (2) Beckmann, P. A. A Review of Polytypism in Lead Iodide. *Cryst. Res. Technol.* **2010**, 45 (5), 455–460.
- (3) Park, H. J.; Ryu, G. H.; Lee, Z. Hole Defects on Two-Dimensional Materials Formed by Electron Beam Irradiation: Toward Nanopore Devices. *Appl. Microsc.* **2015**, 45 (3), 107–114.

Reviewer#2

This manuscript describes the preparation and structural analysis, via electron microscopy, of the 2D transition metal dichalcogenide system PbI₂, a candidate material for optoelectronic applications in the green-to-UV region of the spectrum. The 2D form of this material has received less attention than its multilayered form, in part due to the difficulties of obtaining high quality monolayers. Main results include the preparation of the monolayer (2D) form in a manner amenable to electron microscopy imaging, its adoption of a preferred phase and

orientation relation with the graphene substrate, and the observation of various beam induced defect structures and their evolution over time. I will leave any criticisms warranted of the materials preparation to reviewers better qualified to comment on it, and focus on the electron microscopy.

Though adequate signal-to-noise is a challenge for any thin sample, as evident in the images shown, PbI_2 is well-suited to annular dark field scanning transmission electron microscopy (ADF-STEM) because the strong scaling of the signal with atomic number allows ready discrimination between the Pb and I atoms (with low contribution from the light graphene support), and the structure of the two phases (T-phase and H-phase) are clearly distinguishable by ADF-STEM contrast. Drawing not only on the directly interpretable nature of ADF-STEM for thin samples but also on comparison with electron scattering (multislice) simulations, the authors demonstrate the phase and monolayer nature of their prepared samples and proceed to catalogue a range of (beam-induced) structural defects, many of which evolve dynamically under electron irradiation but some of which, once formed, seem reasonably stable. The figures are generally of high quality and used to good explanatory effect. While this kind of electron microscopy analysis is no longer unusual, the present analysis seems to me generally done well, and the significance of the material should help this work appeal to a wide cross-section of the materials research community. **I would ask the authors to consider the points raised below by way of minor revision before the paper be accepted for publication.**

Main concern:

Reviewer#2 1) Though some motivation is provided in the introduction and conclusion, the discussion of Figs. 4–9 provides a great enumeration of defect structures encountered, but does not convey to me a sense of conceptual framework or practical significance. Especially given that these defects seem to be electron beam induced, I feel the manuscript would be

significantly strengthened if the unity of purpose and significance of these findings could be more clearly conveyed.

Our response: Our work is first of its kind to synthesize pure monolayer PbI_2 using liquid exfoliation method, study the monolayer structure as well as the defect dynamics. We have now sectioned the paper into three parts so as the manuscript can clearly demonstrate the purpose and significance of these findings.

Minor considerations:

The monolayer PbI_2 has not been considered before because of the lack of synthesis method to produce predominantly monolayer structures. Knowing the importance of the monolayer PbI_2 structure and material, this article presents the ways to synthesize and study it. Graphene has played an important role in the other 2D materials and this article presents how graphene can influence and relate to the synthesis and the production of the 2D monolayer PbI_2 as well as help us study its structure and deformation under microscope in its 2D form/. The monolayer study on its structure and defects has never been carried out before and hence, we also present that to the readers for better understanding of the thin layered 2D-crystals. Moreover, this paper presents a detailed analysis on the study of the monolayer PbI_2 which has never been done before, whilst at the same time, contributing to the large scale facile synthesis of it on top of a substrate.

Reviewer#2 2) Line 132: given the time between images shown is many orders of magnitude greater than most atomic dynamics, that holes are seen in Fig. S3 (rather than, say, “thinning”) seems to me to be a less than conclusive demonstrating that the sample is indeed only a monolayer thick. Could the authors please elaborate on why holes would be expected to open up layer-wise?

Our response: Layer by layer opening of holes in few-layered 2D materials under electron beam irradiation is a well-documented phenomenon and relates to the difference in displacement energy for atoms at the surface compared to within the bulk.⁴⁻⁶ Surface atoms have lower energy for displacement than central bulk atoms, which results in the opening of holes from the surfaces first, before the holes open in the central bulk regions. Once a hole opens up with sufficient size in the surface layer to expose the next layer beneath, it then also becomes a surface layer and holes begin to open up in this layer too. This gives the layer by layer hole expansion process for few layered 2D crystals where the first hole opening up leads to visible lattice within the hole and not vacuum. In monolayer 2d crystals, hole opening leads immediately to vacuum. We have also updated the main text for more clarity.

References:

- (4) Zhou, S.; Wang, S.; Li, H.; Xu, W.; Gong, C.; Grossman, C.; Warner, J. H. Atomic Structure and Dynamics of Defects in 2D MoS₂ Bilayers. **2017**.
- (5) Wang, S.; Robertson, A.; Warner, J. H.; Robertson, A. Chem Soc Rev Atomic Structure of Defects and Dopants in 2D Layered Transition Metal Dichalcogenides. **2018**.
- (6) J. H. Warner, M. H. Rummeli, L. Ge, T. Gemming, B. Montanari, N. M. Harrison, B. Büchner, G. A. D. Briggs, Structural transformations in graphene studied with high spatial and fast temporal resolution, Nature Nanotechnology, 4, 500 (2009)

Reviewer#2 3) Line 168–170: “The smaller the difference in the relative lattice spacings between the two crystals, the larger the van der Waals interaction is likely to be. The best lattice match occurs when the PbI₂ adopts 1H phase and is aligned to the arm-chair direction, which agrees with our experimental findings.” Please consider spelling out this connection in greater detail in reference to Fig. 3(s-v), especially since the green lines indicating the “minimum

distance where Pb atom overlaps a C-C bond in graphene, indicating loss of commensuration” seem to me to be of comparable length between Figs. 3(s) and (v).

Our response: The figure is now moved to SI as figure S11. The figure is used to show the distance to a position of commensuration between the Pb atom and the underlying graphene lattice. The green line for the 1H:Arm-chair case is more than twice the distance than all other cases. This helps to understand why it might have a preference for alignment. We have also now added a more definitive DFT calculation which shows the armchair PbI_2 has local energy minima when aligned to the armchair or zig-zag graphene direction s(Figure. 3 (s-t)). This has been described in more details in the figure and the main text now.

Reviewer#2 4) *In Fig. 4(j-l), what was used as the basis of relative alignment? Given the mobility of defects asserted elsewhere, it is not entirely obvious to me that the orange lines on those three successive images correspond to the same set of atoms.

Our response: We have now made a new figure in S.I. 13, highlighting the relative alignment spots. The figure in the paper (Figure 4(j-l)), has been cropped out of this series so that easier to see the edge dynamics of the successive images that correspond to the same set of atoms.

Reviewer#2 5) *The weak contrast at the centre of the defect in Fig. 5(b) is interpreted to indicate a single vacancy (rather than two I vacancies one above the other). However, there seems to me to be similar remnant contrast present in the centre in Fig. 5(b), (j) and (k) but those cases are interpreted as complete vacancies. Please elaborate on how the interpretation / assignment of these experimental images to particular simulated configurations was made.

Our response: The interpretation and assignment of the experimental images to their simulated configurations were made on the line profile of the defects after normalizing their intensities. We have now added the extra information in S.I. 14 on how we have calculated the single

vacancy or complete vacancies of I atoms. During image acquisition, some vacancies will be produced during the imaging process and therefore some residual contrast may appear for some cases, and also during imaging an existing vacancy may be filled during the image acquisition, which also leads to some residue contrast. To gain a full understanding, we measured many point vacancies and draw conclusions from multiple measurements of the same structures. Furthermore, we believe that if the contrast pattern is stable and repeated during multiple sequential images, then it represents a stable configuration of a defect, rather than a transition state. Capturing a structural transition in a vacancy defect can lead to unusual contrast patterns that could be misinterpreted, but these contrast patterns are rarely stable for multiple sequential images because the transition pathways in vacancy changes are rarely between just two stable states, they often involve multiple pathways and movements to nearby lattice sites.

Reviewer#2 6) *In Figs. 8(f-h), (n-p), what is the significance of the red arrows? It seems to imply a direct this-atom-moved-from-here-to-here interpretation that I would consider questionable given the time frame of atomic dynamics relative to the time elapsed between images.

Our response: Yes, it is not possible to tell the pathway of movement of the atoms given the time elapsed between capturing two consecutive images. TEM captures the positions of fixed stability. It is similar to stroboscopic imaging of fast dynamics. Every movie/video has frames that capture still images of a moving event. The red arrows indicate the possibility of migration of the atoms from one position to the other. The electron beam damages the material and forms the nanopore but at the same time, we are able to capture the fast migration of atoms and the self-healing process in the material in real time. However, prolonged exposure to the electron beam ultimately leads excessive sputtering of atoms than the self-healing itself leading to a

bigger nanopore formation. We have now added more text to the main text to explain the figure more properly.

Reviewer#2 7) *In the supplementary material, I find the discussion comparing what Fig. S4 shows to what Fig. S5 shows confusing, especially given the differences in format of the two figures. Please clarify.

Our response: The figure in S.I.7 [Previously S.I.4] shows two different areas of graphene, far apart from each other. This can be seen in figure S7(a-c) and S7(d-f). And yet another example of this is shown in figure S8 where all the flakes are from the same region and hence, all of them have the same lattice orientation. We have now clarified this more in the supplementary text and the formatting has also been changed so that it is easier to interpret.

Reviewer#2 8) *In section S5, I find it hard to make the connections between Figs. S5(c) and (d) on the basis of relative column intensities, though the intercolumn spacing is persuasive. Should this be explicitly stated?

Our response: Thank you for the comment. We have now made changes the S.I. 9 [Previously S.I.5 text to explicitly state it.

Reviewer#2 9) *I also find it hard to see Fig. S6(c) as being the “high resolution image of the area boxed in red color in figure a” – not least, the extent of the fields of view appear to be quite different. Please confirm the correct image has been provided.

Our response: Thank you for pointing this out. We have now marked the image with the correct indicators and the right figure. The image provided in Fig. S10(c) [Previously, Fig. S6(c)] is now, of the same boxed region from Fig. S10(a).

Trivial points:

Reviewer#2 10) *In line 222, “The red arrow indicates the ejection of the electrons...” should presumably refer to the ejection of atoms? (At the risk of being especially pedantic, I’d also argue that what is shown in Fig. 4(i) is a red line, not an arrow since it lacks a head.)

Our response: Thank you for pointing this out. Yes, the red arrow indicated the ejection of the electrons. We have now corrected this in the main text as well as the red arrow.

Reviewer#2 11) *In line 380: “Sigma Aldrick” should read “Sigma Aldrich”.

Our response: We have now corrected this, thank you.

Reviewer#2 12) *In the supplementary material, there are currently two figures both labelled Fig. S5.

Our response: We have now corrected this, thank you.

Reviewer #3: The authors for the first time reported the synthesis of 2D monolayer Lead Iodide nanodisk with diameter of 30-40 nm using liquid-exfoliation method. Then interface it with graphene to form van der Waals heterostructure which may be important for future applications in opto-electronics. Defective dynamic in 2D PbI_2 is also studied. The topic is interesting. All the experimental synthesis and characterization are basically sound. I would recommend it publishing in Nature Communications after some minor revisions.

Reviewer#3 1) The band gap of PbI_2 is quite large. The Most practical applications should be discussed.

Our response: Thank you for the useful comment. The band gap of PbI_2 is 2.4 eV in its bulk form, whereas its 2D monolayer has an indirect bandgap of about 2.5 eV, with possibilities to tune the band gap between 1-3eV. This enables it to PbI_2 to be frequently used for fabrication

of organic-inorganic halide perovskite solar cells, and as a high-energy photon detector material for gamma-rays and X-rays. We have added more on this in the main text.

Reviewer#3 2) Many defects have been formed and observed. Which has the lowest formation energy?

Our response: The lowest formation energy is that of the single iodide vacancy that takes about 3.15 eV to get sputtered out of the system. We have now calculated it using DFT and added it to the main text as well as S.I.14.

Reviewer #4

In this work, Jamie H. Warner and co-workers report the liquid-exfoliation of PbI₂ nanodisks and

systemically study the interaction between PbI₂ and graphene. The fundamental atomic structure and

different defects as well as the edge states under electron beam irradiation are well studied. The method is simple, and the solid experiment has been conducted by ADF-STEM. The interaction between graphene and PbI₂ is illustrated. Some comments, especially the detailed structure information, should be addressed before its acceptance.

Reviewer#4 1) The authors claim that only 2 flakes with bilayer structure are found among more than 300 flakes. However, in figure 1c, 1e and figure s1 and s2, there are small flakes showing thickness is above bilayer.

Our response: They are small particles coming from graphene and not the PbI₂ itself. We have added extra text in the main text and more information in S.I. 4.

Reviewer#4 2) The authors use the supernatant of the dispersion for preparing TEM samples. Where (bottom or top) is the dispersion taken from the container? Do the flakes have the similar thickness in the bottom or top of the container?

Our response: We have used the supernatant (top) of the dispersion for preparing TEM samples. The flakes have different thickness in the bottom and the top of the container. We have added extra text in the main text and more information in S.I. 2.

Reviewer#4 3) The authors should consider all the few-layer PbI_2 not only the monolayer and bilayer.

Our response: The other few-layer PbI_2 is out of scope of this paper.

Reviewer#4 4) Meanwhile, the experiment conditions of liquid-exfoliation for the 300 flakes should be provided in the manuscript.

Our response: We have added more details to the experimental section in the main text.

Reviewer#4 5) Why the PbI_2 prefers the orientation of zigzag aligned to graphene armchair direction rather than the graphene zigzag direction. The detailed reason should be discussed and offered in the manuscript.

Our response: We have now done some DFT calculation on the energy and stability of the PbI_2 flakes on graphene. PbI_2 prefers the orientation of zigzag aligned to graphene armchair direction because that gives the smallest strain in PbI_2 layer. We have now added more detailed explanation in the main text and also modified the figure 3 in the main text to reflect the new addition.

Reviewer#4 6) The authors claim that the best lattice matching occurs in the 1H-PbI₂ with direction aligning to the graphene arm-chair. On the contrary, what's the structure of PbI₂ (1H or 1T) when its direction aligning to the graphene zigzag?

Our response: As we have shown in the results section as well as now with DFT calculations, the PbI₂ does not align to the graphene zigzag direction.

Reviewer#4 7) What's the graphene direction when the edges of PbI₂ are etched to form sharp zig-zag faceted terminations after electron beam irradiation?

Our response: The graphene still maintains its armchair direction even when PbI₂ is being etched to form the sharp zig-zag faceted terminations after electron beam irradiation. We have conducted new experiments to show this phenomenon and have added the information in the main text and the details of the result in S.I. 11.

Reviewer#4 8) Different defects are induced by electron beam irradiation such as 4-membered ring configuration, atoms missing. As for the defects, the significant issue is the stability. Do these structures can be stable in air?

Our response: We have conducted all our experiments under vacuum since that's how we can get the microscope working. However, multilayer PbI₂ is known to exhibit large number of stable vacancies, as discussed in the introduction and the conclusion section of the paper. Any defect in a 2D material that is exposed to air gets functionalized. This changes the nature of the defect from its intrinsic fundamental state, into an oxidized or functionalized version. Defects attract hydrocarbons to locally bind to them, which is why grain boundaries are often dirty compared to the rest of area of graphene growth by CVD. Studies of air functionalized defects is a separate study on its own. There are many fields of research such as batteries, where

exposure to air is not done. Also there are many 2D crystals that are not stable in air, and gloveboxes are used to handle them to study their properties.

Reviewer#4 9) What's the effect of the accumulation of the atoms in Fig (i-j). To form a new material or only a new structure of PbI_2 ? It should be clarified.

Our response: Iodine atoms are very light and they have very low sputtering energy (as calculated from DFT and added to the main text and S.I.) and hence, they get sputtered out from the system very easily. Also looking at the contrast, the single atoms imaged are that of Lead. The atoms aggregate to form clusters or get dispersed on the surface of graphene. We have added more results in S.I. 15. These are usually amorphous and have no special crystal structure.

REVIEWERS' COMMENTS:

Reviewer #1 (Remarks to the Author):

The authors have revised the manuscript based on all reviewers' comments. The novelty and importance of the paper have now been greatly improved. DFT simulation has been used to support experimental observations such as orientation relationship between graphene and PbI₂, and by providing formation energies of point defect and energy barrier for diffusion of atoms. Also, the characterization is now more quantitatively. Instead of merely reporting TEM images in previous version, the paper now provides a comprehensive investigation on this novel 2D materials. I therefore recommend acceptance of the paper.

Reviewer #2 (Remarks to the Author):

The authors have addressed the points I raised. I recommend the manuscript for publication in Nature Communications.

Trivial points:

- * "...by unzipping the outer most atoms to retain its uniform zig-zag termination (Figure 4e)." Should this refer to 4j?
- * "Between figure 8s-p, the top left section..." Should this be figure 8s-t?
- * "This indicates that the area in the red box in figure S5(a) is indeed a monolayer..." Fig. 5 doesn't have a red box. Not sure what this is referring to. Maybe Fig. 7a?
- * "Figure S12(b-c) show the time lapse series of ADF-STEM..." Should be figure S12 (d-e)?
- * Figure S12 caption "...the ejection of the electrons..." should be ejection of atoms?

Reviewer #3 (Remarks to the Author):

I have carefully read the response letter. Basically all the comments raised in the first round have been addressed. I would recommend it publishing in Nature Communications.

Reviewer #4 (Remarks to the Author):

The authors have considerably revised their work and addressed some important issues. Nevertheless, there are a number of specific points I would like the authors to address before a final decision can be reached.

- The main text contains too much information, which may dilute the significance and novelty of the work. The conciseness should be improved by only keeping the most significant parts in the main text.

- Minor concerns:

- a) Figure 1g is not mentioned in the manuscript at all.
- b) Some grammar mistakes, e.g. "iodine atoms" instead of "iodide atoms", "the agglomeration of vacancies leads to a hole formation" etc.
- c) In the first paragraph of part II, "1c and 1g" should be "2c and 2g" based on the context.
- d) Some letters of figures are not visible, e.g. Figure 3s.
- e) Both "Figure" and "figure" are used throughout the text. Please make it consistent.

Reviewer #1

The authors have revised the manuscript based on all reviewers' comments. The novelty and importance of the paper have now been greatly improved. DFT simulation has been used to support experimental observations such as orientation relationship between graphene and PbI₂, and by providing formation energies of point defect and energy barrier for diffusion of atoms. Also, the characterization is now more quantitatively. Instead of merely reporting TEM images in previous version, the paper now provides a comprehensive investigation on this novel 2D materials. I therefore recommend acceptance of the paper.

Reviewer #2

The authors have addressed the points I raised. I recommend the manuscript for publication in Nature Communications.

Trivial points:

Reviewer#2 1) "...by unzipping the outer most atoms to retain its uniform zig-zag termination (Figure 4e)." Should this refer to 4j?

Our Response: This should refer to 4i. We have now edited it.

Reviewer#2 2) "Between figure 8s-p, the top left section..." Should this be figure 8s-t?

Our Response: Yes, and we have now changed it figure 8s-t.

Reviewer#2 3) "This indicates that the area in the red box in figure S5(a) is indeed a monolayer..." Fig. 5 doesn't have a red box. Not sure what this is referring to. Maybe Fig. 7a?

Our Response: We have now removed the 'red box', since the sentence refers to all of figure S5(a).

Reviewer#2 4) "Figure S12(b-c) show the time lapse series of ADF-STEM..." Should be figure S12 (d-e)?

Our Response: Yes, it has now been edited to read figure S12 (d-e).

Reviewer#2 5) Figure S12 caption "...the ejection of the electrons..." should be ejection of atoms?

Our Response: Yes, it has now been edited to read "ejection of atoms".

Reviewer #3

I have carefully read the response letter. Basically all the comments raised in the first round have been addressed. I would recommend it publishing in Nature Communications.

Reviewer #4

The authors have considerably revised their work and addressed some important issues. Nevertheless, there are a number of specific points I would like the authors to address before a final decision can be reached.

Reviewer#4 1) The main text contains too much information, which may dilute the significance and novelty of the work. The conciseness should be improved by only keeping the most significant parts in the main text.

Our Response: We have now sectioned the work into three sub categories of results, namely (a) Synthesis (b) 1-H structural phase (c) Structural defects. We believe that this would now help make the results section more direct and easy to understand with all the information provided in the main text.

Minor concerns:

Reviewer#4 2) Figure 1g is not mentioned in the manuscript at all.

Our Response: Fig. 1g should have been added in the main text, referring to the dimensions of the exfoliated flakes.

Reviewer#4 3) Some grammar mistakes, e.g. "iodine atoms" instead of "iodide atoms", "the

agglomeration of vacancies leads to a hole formation” etc.

Our Response: We have now corrected the mistakes and also checked the manuscript for other possible grammatical mistakes.

Reviewer#4 4) In the first paragraph of part II, “1c and 1g” should be “2c and 2g” based on the context.

Our Response: It has now been corrected.

Reviewer#4 5) Some letters of figures are not visible, e.g. Figure 3s.

Our Response: We have now edited the figure to make it more visible.

Reviewer#4 6) Both “Figure” and “figure” are used throughout the text. Please make it consistent.

Our Response: We have now changed them all to “Fig. ” to follow the Nature Communications format of writing.

EDITORIAL REQUESTS:

1) Please remove ORCIDs present in author's information.

Response: It has now been removed.

2) Please remove the image present after Reference list.

Response: It has now been removed.

TITLE PAGE

3) Please edit the title so that it is 15 words or fewer and does not include punctuation.

Response: It is now less than 15 words.

4) Please shorten the abstract to 150 words or fewer.

Response: It is now 150 words.

MAIN TEXT

5) The main text should include only the following sections: Introduction, Results, and optional Discussion, each of which must begin with a heading. All other section headings should be removed or renamed. Please remove the subheadings from the Discussion section.

Response: There is now only Introduction, Results and Discussion sections. The other headings in Results section have been renamed.

LANGUAGE AND STYLE

6) Please remove language such as "new", "novel", "for the first time", "unprecedented", etc. Novelty should be clear from the context.

Response: The claims using these languages have now been removed. The actual description of experimental results, such as, "breaking of previous bond and formation of new atomic

bonds, as shown in figure....' has been kept as it is, as it is not claiming new science but only describing the results.

7) Please do not use italics or bold font to convey emphasis (in both the main text and the display items).

Response: Bold and Italic fonts have been removed.

8) Please make sure that mathematical terms throughout your manuscript and Supplementary Information (including in figures, figure axes, and legends) conform strictly to the following guidelines. Equations should be supplied in editable format, and not as images. Scalar variables (e.g. x , V , χ) should be typeset in italic, whereas multi-letter variables should be formatted in roman. Constants (e.g. h , G , c) should be typeset in italics (the only exceptions being e , i , π , which should be typeset in Roman) and vectors (such as r , the wavevector k , or the magnetic field vector B) should be typeset in bold without italics. In contrast, subscripts and superscripts should only be italicised if they too are variables or constants. Those that are labels (such as the 'c' in the critical temperature, T_c , the 'F' in the Fermi energy, E_F , or the 'crit' in the critical current, I_{crit}) should be typeset in roman. To avoid doubt, unit dimensions should be expressed using negative integers

(e.g. $\text{kg m}^{-1} \text{s}^{-2}$, not kg/ms^2) or the word 'per'.

Response: Equations are supplied in editable format.

9) Please italicise scalar variables (except for multi-letter variables). Subscripts of variables, where they are shortened forms of words or phrases, should not be italic. Example: in the critical temperature, T_c , the 'T' should be in italics and the 'c' should be in roman.

Response: Variables in the supplementary information have been italized.

METHODS AND DATA

10) All Nature Communications manuscripts must include a section titled "Data Availability" as a separate section after the Methods section and before the References. For more information on this policy, and a list of examples, please see <http://www.nature.com/authors/policies/data/data-availability-statements-data-citations.pdf>

Response: Data Availability has been added after the Methods section and before the References.

11) DATA SOURCES: Nature Research policies strongly encourage deposition of research data in public repositories and in some cases this is mandatory, and you may have been previously advised if that was the case. If you need help depositing and curating your research data (including raw and processed data, text, video, audio and images) you should consider:

Response: There is no data sources to add otherwise.

END NOTES

12) Please supply an "Author Contributions" section after the Acknowledgement section that refers to all authors.

Response: "Author Contributions" section has been added after the Acknowledgement section that refers to all authors.

13) Please provide a "Competing Interests" section after the "Author Contributions" section that refers to all authors. If there are no competing interests, please add the statement "The authors declare no competing interests."

Response: The statement "The authors declare no competing interests" has been added to the "Competing Interests" section after the "Author Contributions" section.

DISPLAY ITEMS

14) Please check whether your manuscript or Supplementary Information contain third-party images, such as figures from the literature, stock photos, clip art or commercial satellite and map data. We strongly discourage the use or adaptation of previously published images, but if this is unavoidable, please request the necessary rights documentation to re-use such material from the relevant copyright holders and return this to us when you submit your revised manuscript.

Response: We do not have any third-party images in the main text or the supplementary information.

15) Please include a brief title for all figure legends that summarises the whole figure and does not refer to specific panels. Please ensure that every figure panel is described in the legend.

Response: We have included a brief title for figure legends in all figure panels.

16) Please ensure that at least one micrograph in each equivalent group in each figure is supplied with a representative scale bar, whose length is stated in the corresponding figure legend.

Response: All representative scale bars have been supplied, wherever necessary. All the scale bars have their lengths in the figure, otherwise, the lengths have been stated in the figure caption.

17) Some figures in your paper include bar charts. Please overlay the corresponding data points (as dot plots) in the bar charts.

Response: We have one bar chart in figure 1. The data point (Counts) is the y-axis of the bar chart, which is 300 and 2. Hence, we think that there cannot be dot plots in the bar chart.

18) Please define any new abbreviations, symbols or colours present in your figures in the associated legends. Please do not use symbols in your legend, instead please write out the symbols in words (blue circles, red dashed line, etc.).

Response: There is no new abbreviations or symbols present in the figures in the associated legends. The colours have been described in the associated legends of the figure.

SUPPLEMENTARY INFORMATION

19) We do not edit Supplementary Information files; they will be uploaded with the published article as they are submitted with the final version of your manuscript. Any tracked changes should be removed from the file and the file should be provided as a PDF file. Supplementary Figures do not need to be provided separately.

Response: We are providing the finalized version of the Supplementary Information file along with the main text.

20) In the Supplementary Information file and the main manuscript text, supplementary items must be labelled and cited using only the following formats: Supplementary Figure 1, Supplementary Table 1, Supplementary Methods, Supplementary Note 1, Supplementary Discussion, and Supplementary References. Please note the use of "Supplementary" and that we do not use the "S" prefix.

Response: The main text has now been edited to read ‘‘supplementary information’’.

21) Please label supplementary equations sequentially as (1), (2), (3), etc. (and without an "S" prefix).

Response: The supplementary equations in the supplementary information file have now been labelled properly.

22) We encourage increased transparency in peer review by publishing the reviewer comments and author rebuttal letters of our research articles, if the authors agree. Such peer review material is made available as a supplementary peer review file. Please state in the

cover letter 'I wish to participate in transparent peer review' if you want to opt in, or 'I do not wish to participate in transparent peer review' if you don't.

Response: I wish to participate in transparent peer review.

23) An updated editorial policy checklist that verifies compliance with all required editorial policies must be completed and uploaded as a related manuscript file with the revised manuscript. All points on the policy checklist must be addressed; if needed, please revise your manuscript in response to these points. Please note that this form is a dynamic "smart pdf" and must therefore be downloaded and completed in Adobe Reader, instead of opening it in a web browser. Editorial policy checklist:

<https://www.nature.com/authors/policies/Policy.pdf>

Response: We have addressed all the point and responded to it in this cover letter.

24) Your paper will be accompanied by a two-sentence Editor's summary, of between 250-300 characters including spaces, when it is published on our homepage. Could you please approve the draft summary below or provide us with a suitably edited version. "Imaging liquid phase exfoliated nanosheets on suspended graphene via annular dark field STEM can enable identification of various defects, vacancies and their migration. Here, the authors report matching of zig-zag edges of monolayer PbI₂ with graphene arm-chairs leading to a phase shift from 1T to 1H structure to maximize commensuration of the lattices."

Response: We accept this.